# DISCRETE COPULA DIFFUSION

**Anji Liu**[1,2]**, Oliver Broadrick**[1]**,  Mathias Niepert**[2]**, Guy Van den Broeck**[1]

[1]Department of Computer Science, University of California, Los Angeles, USA
[2]Institute for Artificial Intelligence, University of Stuttgart, Germany
`{liuanji,obroadrick,guyvdb}@cs.ucla.edu`
`mathias.niepert@ki.uni-stuttgart.de`

## ABSTRACT

Discrete diffusion models have recently shown significant progress in modeling complex data, such as natural languages and DNA sequences. However, unlike diffusion models for continuous data, which can generate high-quality samples in just a few denoising steps, modern discrete diffusion models still require hundreds or even thousands of denoising steps to perform well. In this paper, we identify a fundamental limitation that prevents discrete diffusion models from achieving strong performance with fewer steps – they fail to capture dependencies between output variables at each denoising step. To address this issue, we provide a formal explanation and introduce a general approach to supplement the missing dependency information by incorporating another deep generative model, termed the *copula* model. Our method does not require fine-tuning either the diffusion model or the copula model, yet it enables high-quality sample generation with significantly fewer denoising steps. When we apply this approach to autoregressive copula models, the combined model outperforms both models individually in unconditional and conditional text generation. Specifically, the hybrid model achieves better (un)conditional text generation using 8 to 32 times fewer denoising steps than the diffusion model alone. In addition to presenting an effective discrete diffusion generation algorithm, this paper emphasizes the importance of modeling inter-variable dependencies in discrete diffusion.[1]

## 1 INTRODUCTION

Discrete diffusion models have recently achieved significant progress in modeling complex data such as natural languages (Campbell et al., 2022; Sahoo et al., 2024), protein sequences (Gruver et al., 2023; Morehead et al., 2023), and graphs (Vignac et al., 2022; Huang et al., 2023). In particular, recent discrete diffusion models for text generation (Lou et al., 2024; Sahoo et al., 2024; Shi et al., 2024) have matched or even surpassed the performance of autoregressive models at the scale of GPT-2 (Radford et al., 2019). Additionally, discrete diffusion models offer improved inference-time controllability using guidance from auxiliary models such as classifiers (Dhariwal & Nichol, 2021), making them suitable for controlled generation tasks (Li et al., 2022; Han et al., 2023).

Despite these promising results, discrete diffusion models still require hundreds to thousands of denoising steps to produce high-quality samples (Austin et al., 2021; Sahoo et al., 2024), significantly affecting their efficiency. In this paper, we identify a fundamental limitation in most discrete diffusion models that hinders their ability to generate high-quality samples in just a few steps.

We illustrate the problem in Figure 1. At each denoising step, a partially completed sample shown in the top-left is fed into a sequence-to-sequence denoising model, which predicts the univariate marginal distributions for each masked token independently. A new output sequence is then sampled based on these univariate marginals before proceeding to the next denoising step. The key issue with this process is that when multiple "edits" (i.e., replacing masked tokens with data tokens) are made simultaneously, the model does not account for the joint probability of these changes occurring together. As a result, the generated samples often lack coherence, as shown in the bottom-

---

[1]Code is available at https://github.com/liuanji/Copula-Diffusion.

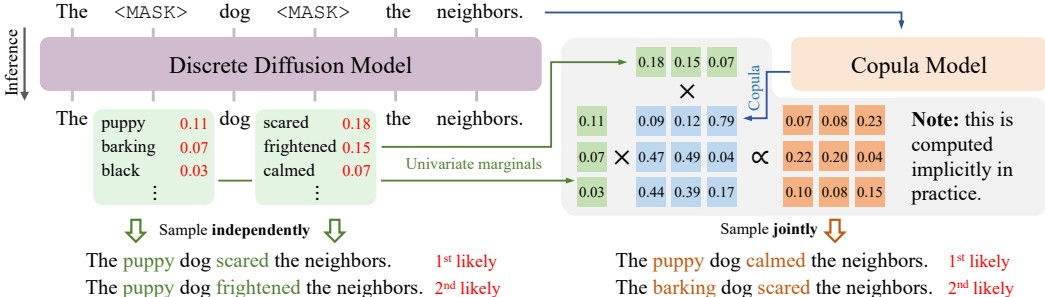

Figure 1: **Discrete Copula Diffusion (DCD).** At each denoising step, a partially completed sequence is given as input (top-left). The diffusion model independently predicts the univariate marginals for each masked token, which leads to the samples in the bottom-left. DCD introduces an additional copula model (top-right) to capture the inter-variable dependencies, thereby supplementing the information missed by the diffusion model. By combining outputs from both models in a principled way, DCD achieves better performance than either model individually (see improved samples in the bottom-right), enabling few-step discrete diffusion generation.

left of Figure 1. This problem is exacerbated in few-step generation, where many tokens must be edited simultaneously. We formally demonstrate that if the diffusion model predicts each variable independently, an irreducible term (in addition to the data entropy) remains in the negative evidence lower bound (ELBO), preventing the model from perfectly capturing the data distribution.

We propose using a generative model, which we refer to as the copula model, to compensate for the missing dependency information between output variables at each denoising step. Our method operates only at inference time and can be adapted to any discrete diffusion model and a wide range of copula models. As illustrated on the right side of Figure 1, the input sequence is also fed into a copula model that (implicitly) produces information on inter-variable dependencies. This information is combined with the univariate marginals predicted by the diffusion model to produce a more accurate distribution, resulting in higher-quality samples, shown in the bottom-right corner.

We formally show that the univariate marginals from the diffusion model and the dependencies captured by the copula model can be combined in a principled way, leading to a better approximation of the true denoising distribution under mild assumptions. Further, finding this combined distribution reduces to solving a convex optimization problem that can be efficiently approximated in practice.

By instantiating the copula model as an autoregressive deep generative model such as GPT (Radford et al., 2019), we propose an algorithm that combines any pretrained discrete diffusion model with an autoregressive model to form a hybrid model called **D**iscrete **C**opula **D**iffusion (DCD). This model is capable of producing high-quality (un)conditional samples with only a few denoising steps. Empirical results on text and antibody generation show that DCD significantly outperforms both of its base models. Moreover, DCD achieves comparable or better performance using 8 to 32 times fewer denoising steps compared to the base discrete diffusion model. In addition to proposing a discrete diffusion model capable of few-step generation, we emphasize the importance of modeling inter-variable dependencies in discrete diffusion models.

## 2 PRELIMINARIES

We aim to model the joint distribution of variables $\mathbf{X}_0$, a set of categorical variables with $C$ categories. Discrete diffusion models (Austin et al., 2021) learn to sample from $p(\mathbf{X}_0)$ by modeling the reversal of the following noising process involving $\mathbf{X}_0$ and a set of auxiliary variables $\{\mathbf{X}_t\}_{t=1}^{T}$:

$$\forall t \in \{1, \dots, T\} \quad q(\boldsymbol{x}_t | \boldsymbol{x}_{t-1}) := \mathrm{Cat}(\boldsymbol{x}_t; Q_t \cdot \boldsymbol{x}_{t-1}), \tag{1}$$

where $\mathrm{Cat}(\boldsymbol{x}; \mathbf{p})$ refers to the Categorical distribution over $\boldsymbol{x}$ with class probabilities $\mathbf{p}$, and $Q_t$ is a $C \times C$ transition matrix that is applied independently to every variable $x_{t-1}^i$ (denote $x_{t-1}^i$ as the $i$th variable of $\boldsymbol{x}_{t-1}$) to get the corresponding categorical distribution of $x_t^i$. Specifically, each variable $x_{t-1}^i$ is treated as a one-hot vector of size $C \times 1$, which is then multiplied by $Q_t$ to compute the class probabilities of $x_t^i$. The noising process is designed such that $p(\boldsymbol{x}_T)$ follows a simple distribution regardless of the data distribution.

Instead of using a fixed number of predefined time steps, we can treat $t$ as a continuous variable within the range $[0, T]$. The noising process is now defined by the rate of change of $p(\boldsymbol{x}_t)$ w.r.t. $t$: $\frac{dp(\boldsymbol{x}_t)}{dt} = Q \cdot p(\boldsymbol{x}_t)$, where $Q \in \mathbb{R}^{C \times C}$ is a transition rate matrix. For any $0 \le s < t \le T$, we have

$$q(\boldsymbol{x}_t | \boldsymbol{x}_s) := \mathrm{Cat}(\boldsymbol{x}_t; \exp((t-s) \cdot Q) \cdot \boldsymbol{x}_s),$$

where $\exp(\cdot)$ denotes the matrix exponential.

Discrete diffusion models represent the reverse diffusion process as a Markov chain from $\boldsymbol{x}_T$ to $\boldsymbol{x}_0$, effectively reversing the noising process. Specifically, the reverse diffusion is modeled as:

$$p_\theta(\boldsymbol{x}_{0:T}) := p(\boldsymbol{x}_T) \prod_{t=0}^{T-1} p_\theta(\boldsymbol{x}_t | \boldsymbol{x}_{t+1}).$$

In the discrete-time framework, the model is trained by maximizing the ELBO, which is defined by the forward joint distribution $(q(\boldsymbol{x}_{1:T} | \boldsymbol{x}_0) p(\boldsymbol{x}_0))$ and the reverse joint distribution $(p_\theta(\boldsymbol{x}_{0:T}))$ (Ho et al., 2020). In the continuous-time framework, we can either adopt an extended ELBO objective (Zhao et al., 2024) or to learn the likelihood ratios $\{p(\boldsymbol{x}_t')/p(\boldsymbol{x}_t)\}_{\boldsymbol{x}_t, \boldsymbol{x}_t'}$, allowing for the recovery of $p(\boldsymbol{x}_s | \boldsymbol{x}_t)$ $(s < t)$ in an indirect manner (Lou et al., 2024; Meng et al., 2022; Sun et al., 2022). Following the reverse diffusion process, sampling from a diffusion model involves first sampling from the prior $p(\boldsymbol{x}_T)$ and then recursively sampling $\boldsymbol{x}_{T-1}, \dots, \boldsymbol{x}_0$ following $\{p_\theta(\boldsymbol{x}_t | \boldsymbol{x}_{t-1})\}_{t=0}^{T-1}$.

## 3 CHALLENGE OF MODELING VARIABLE DEPENDENCIES

Unlike continuous diffusion models, which can produce high-quality samples with just a few steps (e.g., Song et al. (2023); Zhou et al. (2024)), discrete diffusion models exhibit a strong positive correlation between sample quality and the number of denoising steps. For instance, to generate 1024 text tokens, a recent discrete diffusion model SEDD (Lou et al., 2024) requires 1024 steps to reach around 35 perplexity (PPL), while with 32 denoising steps the PPL is only around 130.

We argue that the need for a large number of sampling steps in discrete diffusion models stems from their inability to capture inter-variable dependence among the outputs. Specifically, at each time step $t$, discrete diffusion models independently sample each variable from $\boldsymbol{x}_t$ conditioned on $\boldsymbol{x}_{t+1}$, i.e., $p(\boldsymbol{x}_t | \boldsymbol{x}_{t+1}) := \prod_i p(x_t^i | \boldsymbol{x}_{t+1})$. As a result, when changing multiple variables from $\boldsymbol{x}_{t+1}$ to $\boldsymbol{x}_t$, the model fails to account for the joint probability of these modifications happening together. In the following, we first quantitatively analyze the performance degradation caused by this independent denoising assumption. We then discuss approaches to mitigate this issue.

**Quantifying the Performance Drop.** The total correlation of a distribution $p(\mathbf{X})$ is the KL-divergence between itself and the product of its univariate marginals:

$$\mathrm{D_{TC}}(p(\mathbf{X})) := \sum_{\boldsymbol{x}} p(\boldsymbol{x}) \log\left(p(\boldsymbol{x}) / \prod_i p(x_i)\right).$$

The following result demonstrates that, under the independent denoising assumption, there is an irreducible component in the ELBO that directly stems from ignoring inter-variable dependencies.

**Proposition 1.** *Assume the denoising distributions $\{p_\theta(\boldsymbol{x}_t | \boldsymbol{x}_{t+1})\}_{t=0}^{T-1}$ are fully factorized. Let $\mathrm{H}(p(\mathbf{X}))$ denote the entropy of $p(\mathbf{X})$. For any choice of denoising distributions (or equivalently, any parameterization $\theta$), the negative ELBO of the diffusion model is lower bounded by*

$$\mathrm{H}(p(\mathbf{X}_0)) + \sum_{t=1}^{T} \mathrm{D_{TC}}(q(\mathbf{X}_{t-1} | \mathbf{X}_t)), \quad \text{where } \mathrm{D_{TC}}(p(\mathbf{Y}|\mathbf{X})) := \mathbb{E}_{\boldsymbol{x} \sim p}\left[\mathrm{D_{TC}}(p(\mathbf{Y}|\boldsymbol{x}))\right]. \quad (2)$$

The first term represents the entropy of the data distribution and is irreducible. The second term additionally depends on the noising process and the chosen noise levels, which set an upper limit on the performance of discrete diffusion models that use the independent denoising assumption. Note that although $\mathrm{D_{TC}}(q(\mathbf{X}_t | \mathbf{X}_{t-1}))$ is zero according to the definition of the noising process, $\mathrm{D_{TC}}(q(\mathbf{X}_{t-1} | \mathbf{X}_t))$ is not unless the data distribution is fully factorized.

**Closing the Performance Gap.** While increasing the number of denoising steps can improve sample quality, it also introduces significant computational overhead during inference. Our goal is to

use fewer denoising steps while maintaining good sample quality. As shown in Proposition 1, given a fixed noising strategy and the number of denoising steps, the only way to reduce the negative ELBO lower bound in Equation (2) is to relax the independent denoising assumption. That is, in addition to modeling the univariate marginals, we must also account for dependencies between variables.

The challenge of capturing inter-variable dependencies during each denoising step can be addressed through adjustments during either training or inference. A direct approach involves modeling both the univariate marginals and the inter-variable dependencies within the diffusion model. However, this requires improving existing sequence-to-sequence architectures (e.g., Devlin (2018)) to capture dependencies between output variables directly, which is not very well studied in the literature.

Instead, we propose an inference-time solution that complements the information missed by the pre-trained discrete diffusion model. Specifically, we aim to combine the univariate marginals produced by the diffusion model with the inter-variable dependencies learned by another (possibly smaller) deep generative model, which we refer to as the *copula model*. The term "copula" traditionally refers to the dependencies between random variables in statistics (Nelsen, 2006).

## 4 Modeling Variable Dependencies with Copula Models

As motivated in the previous section, our main goal is to combine the univariate marginals produced by the diffusion model with the inter-variable dependencies captured by a copula model. In this section, we first formalize the concept of "combining" two such distributions in a general context (Sec. 4.1). We then specialize the formulation to the case of diffusion models (Sec. 4.2).

### 4.1 Combining Univariate Marginals with Inter-Variable Dependencies

In this section, we discuss how to best inject inter-variable dependence using copula models given a target distribution $p_{\text{tar}}$ over $\mathbf{X}$. Assume we have access to $p_{\text{tar}}$ through two sources: (i) the set of all univariate marginal distributions $\{p_{\text{tar}}(X_i)\}_i$, and (ii) an estimate $p_{\text{est}}$ of the target distribution coming from the copula model, which is also a generative model. Our goal is to combine these two estimates to construct $\hat{p}$ that is "closer" to the true distribution $p_{\text{tar}}$ than either estimate individually.

We construct $\hat{p}$ as the distribution that (i) matches the set of univariate marginals $\{p_{\text{tar}}(X_i)\}_i$, and (ii) minimizes the KL divergence to $p_{\text{est}}$. The intuition is that by ensuring $\hat{p}$ has the correct univariate marginals, we can achieve a good approximation of $p_{\text{tar}}$ even if $p_{\text{est}}$ is biased. To formalize this, we first define information projection (I-projection).

**Definition 1.** The I-projection of a distribution $q(\mathbf{X})$ onto a set of distributions $\mathcal{P}$ over $\mathbf{X}$ is
$$p^* = \arg\min_{p \in \mathcal{P}} \mathrm{D}_{\mathrm{KL}}(p \,\|\, q).$$

Let $\mathcal{P}^p_{\text{mar}}$ denote the set of distributions over $\mathbf{X}$ that share the same univariate marginals as $p$. We define $\hat{p}$ as the I-projection of $p_{\text{est}}$ onto $\mathcal{P}^{p_{\text{tar}}}_{\text{mar}}$. The following proposition shows that regardless of the initial estimate $p_{\text{est}}$ of $p_{\text{tar}}$, the I-projection $\hat{p}$ will be an improved estimate of $p_{\text{tar}}$ in KL-divergence.

**Proposition 2.** *If there exists $i$ and $x_i$ s.t. $p_{\text{tar}}(x_i) \neq p_{\text{est}}(x_i)$, then $\mathrm{D}_{\mathrm{KL}}(p_{\text{tar}} \,\|\, \hat{p}) < \mathrm{D}_{\mathrm{KL}}(p_{\text{tar}} \,\|\, p_{\text{est}})$.*

Having now seen that $\hat{p}$ is an improved estimate of $p_{\text{tar}}$, we next explore whether it is feasible to compute $\hat{p}$ given $\{p_{\text{tar}}(X_i)\}_i$ and $p_{\text{est}}$. We start by showing that $\hat{p}$ has a simple form.

**Proposition 3.** *Assume $\forall \boldsymbol{x}, p_{\text{tar}}(\boldsymbol{x}) > 0$ and $p_{\text{est}}(\boldsymbol{x}) > 0$. Then $\hat{p}$ exists and has the form*
$$\hat{p}(\boldsymbol{x}) = p_{\text{est}}(\boldsymbol{x}) \cdot \prod_i \sigma_i(x_i),$$
*where $\sigma_i$ is a positive function that depends on $x_i$.*

Assume $\mathbf{X}$ consists of $N$ categorical variables, each with $C$ categories, we can represent the factors $\{\sigma_i\}_i$ using a matrix $\mathbf{V} \in \mathbb{R}^{N \times C}$. Under this representation, the combined distribution is
$$\hat{p}(\boldsymbol{x}) = p_{\text{est}}(\boldsymbol{x}) \cdot \prod_i \exp(\mathbf{V}[i, x_i]), \tag{3}$$
where $\mathbf{V}[i, j]$ denotes the element at the $i$th row and $j$th column of $\mathbf{V}$ and $\mathbf{V}[i, x_i] = \log \sigma_i(x_i)$. Determining the true matrix $\mathbf{V}^*$ corresponding to $\hat{p}$, which is the I-projection of $p_{\text{est}}$ onto $\mathcal{P}^{p_{\text{tar}}}_{\text{mar}}$, can be reformulated as solving the following convex optimization problem.

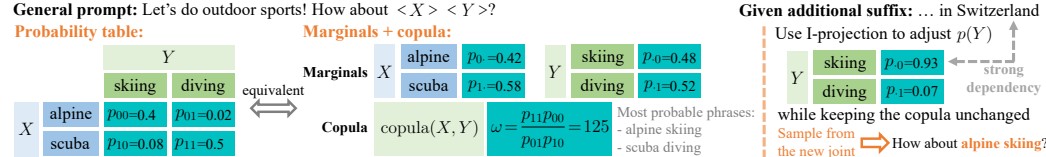

Figure 2: Illustration of the decomposition of a distribution into univariate marginals and a copula.

**Theorem 1.** *If $\mathbf{V}^*$ minimizes the following convex objective function, then the corresponding $\hat{p}$ defined by Equation (3) is the I-projection of $p_{\text{est}}$ onto $\mathcal{P}_{\text{mar}}^{p_{\text{tar}}}$.[2]*

$$\mathcal{L}(\mathbf{V}; p_{\text{tar}}, p_{\text{est}}) := \sum_{\boldsymbol{x}} p_{\text{est}}(\boldsymbol{x}) \cdot \prod_i \exp(\mathbf{V}[i, x_i]) - \sum_{i=1}^{N} \sum_{x_i=1}^{C} \mathbf{V}[i, x_i] \cdot p_{\text{tar}}(x_i). \qquad (4)$$

We proceed to explain why I-projecting $p_{\text{est}}$ leads to a better estimate of $p_{\text{tar}}$ as suggested by Proposition 2. In general, a joint distribution can be viewed as combining two independent pieces of information: (i) a set of univariate marginal distributions and (ii) a *copula* describing the association or dependence among the variables. By the classical work of Sklar (1959), for continuous variables the copula can take the form of a joint distribution with uniform margins and can be combined quite simply with univariate marginal distributions to recover the full joint distribution, a fact heavily exploited in statistics (Nelsen, 2006). While the discrete case is somewhat less straightforward, recent work of Geenens (2020) has developed the fundamental notions of discrete copula modeling as well, where the information of a copula can be parameterized by odds ratios.

Figure 2 shows an example consisting of two binary variables $X$ and $Y$. The probability table on the left can be equivalently expressed using univariate marginals (i.e., $p_{0\cdot}, p_{1\cdot}, p_{\cdot0}, p_{\cdot1}$) and the odds ratio (i.e., copula) $\omega := \frac{p_{00}p_{11}}{p_{01}p_{10}}$ as shown in the middle of Figure 2. Intuitively, $\omega = 125$ indicates that the phrases "alpine skiing" and "scuba diving" are more likely than others (e.g., "alpine diving"), and the marginals decide which of the two phrases appears more frequently. The idea of representing the copula with odds ratios generalizes to the multivariate case and is presented in Appendix C.

The following result demonstrates that, under its functional form in Equation (3), I-projecting $p_{\text{est}}$ onto $\mathcal{P}_{\text{mar}}^{p_{\text{tar}}}$ only improves the univariate marginals and leaves the copula unchanged regardless of $\mathbf{V}$.

**Proposition 4.** *For a positive distribution $p$ and any $\mathbf{V} \in \mathbb{R}^{N \times C}$, the distribution $q(\boldsymbol{x}) \propto p(\boldsymbol{x}) \cdot \prod_i \exp(\mathbf{V}[i, x_i])$ has the same copula as $p$.*

In general, Proposition 4 holds because scaling factors (e.g., $\exp(\mathbf{V}[i, x_i])$) cancel in odds ratios. For example, in the $2 \times 2$ case in Figure 2, scaling the top row of the probability table by $a$ would result in the odds ratio $\omega = \frac{a p_{00} p_{11}}{a p_{01} p_{10}} = \frac{p_{00} p_{11}}{p_{01} p_{10}}$.

## 4.2 MODELING DEPENDENCE IN DISCRETE DIFFUSION MODELS

Recall from Section 3 that our goal is to capture inter-variable dependencies between the output variables at each denoising step (e.g., sampling $\boldsymbol{x}_t$ from $q(\mathbf{X}_t | \boldsymbol{x}_{t+1})$). Similar to the general case shown in Section 4.1, we first have a set of univariate marginals $\{p_{\text{dm}}(X_t^i | \boldsymbol{x}_{t+1})\}_i$ from the diffusion model. Notably, these univariate marginals are fairly accurate since for both discrete-time and continuous-time diffusion models, if their respective training losses are minimized, the model recovers the true univariate marginals. This is formally justified in Appendix D.

Alongside the univariate marginals, we assume access to a copula model that encodes a distribution over $\mathbf{X}_t$. Following Section 4.1, combining the copula model's distribution with the univariate marginals from the diffusion model will lead to an improved estimate of $q(\mathbf{X}_t | \boldsymbol{x}_{t+1})$ (Prop. 2).

The performance of the augmented diffusion model hinges on two key questions: (i) how well can the copula model capture the inter-variable dependencies in $q(\mathbf{X}_t | \boldsymbol{x}_{t+1})$ (defined by the data distribution and the noising process); (ii) given a good copula distribution, how to effectively combine it with the univariate marginals obtained from the diffusion model, i.e., how to solve Equation (4).

---

[2]Equation (4) closely resembles the matrix scaling problem (Idel, 2016). See Appendix B for details.

## 5 Autoregressive Models as Copula Models

This section answers the two questions above tailored to the case where the copula model is an autoregressive model such as GPT (Radford et al., 2019) and State Space Models (Dao & Gu, 2024). Specifically, Section 5.1 discusses how to approximate $q(\mathbf{X}_t|\boldsymbol{x}_{t+1})$ using an autoregressive model trained on the clean data distribution $p(\mathbf{X}_0)$ under certain noising processes. Section 5.2 explores the process of performing I-projection from the (autoregressive) copula distribution onto the set of distributions with univariate marginals $\{p_{\mathrm{dm}}(X_t^i|\boldsymbol{x}_{t+1})\}_i$. Finally, Section 5.3 summarizes the sampling procedure with a discrete diffusion model and an autoregressive copula model.

### 5.1 Extracting Copula Distributions from Autoregressive Models

At step $t$, to sample $\boldsymbol{x}_t$ conditioned on $\boldsymbol{x}_{t+1}$, we need a copula distribution $p_{\mathrm{copula}}(\mathbf{X}_t)$ that closely approximates $q(\mathbf{X}_t|\boldsymbol{x}_{t+1})$. While this might suggest that the copula model should also be trained with a diffusion model objective, which brings us back to the problem of modeling inter-variable dependencies, we show that any model trained on the clean data distribution can serve as a copula model that indirectly approximates $q(\mathbf{X}_t|\boldsymbol{x}_{t+1})$ under the absorbing mask forward noising process.

The absorbing mask noising process gradually converts data tokens in $\boldsymbol{x}_0 \sim p(\mathbf{X}_0)$ to a new category denoted <MASK> through the sequence $\boldsymbol{x}_1, \ldots, \boldsymbol{x}_T$. Specifically, each token in $\boldsymbol{x}_0$ is independently converted to <MASK> with probabilities $0 < \alpha_1 < \ldots < \alpha_T = 1$ in $\boldsymbol{x}_1, \ldots, \boldsymbol{x}_T$, respectively. This is a widely used noising strategy for discrete diffusion models. Since this process only transforms data tokens into the mask token, it preserves the dependencies between the remaining unmasked tokens. Therefore, we can decompose $q(\mathbf{X}_t|\boldsymbol{x}_{t+1})$ as $q(\boldsymbol{x}_t|\boldsymbol{x}_{t+1}) = \sum_{\tilde{\boldsymbol{x}}_t} q(\tilde{\boldsymbol{x}}_t|\boldsymbol{x}_{t+1})q(\boldsymbol{x}_t|\tilde{\boldsymbol{x}}_t, \boldsymbol{x}_{t+1})$, where $q(\tilde{\boldsymbol{x}}_t|\boldsymbol{x}_{t+1})$ is inuitively capturing the joint distribution of generating all currently masked tokens, and $q(\boldsymbol{x}_t|\tilde{\boldsymbol{x}}_t, \boldsymbol{x}_{t+1})$ captures only the choice of which currently masked tokens will actually be generated. Formally, define $I$ as the set of variables $i$ such that $x_{t+1}^i = $ <MASK> and $J$ as its complement. The auxiliary distributions have the following form.

**Proposition 5.** *Assume $p(\mathbf{X}_0)$ is the clean data distribution and $\{q(\mathbf{X}_t|\boldsymbol{x}_{t-1})\}_{t=1}^T$ follows the absorbing mask noising process. Let $\alpha_t$ be the probability of conversion to the mask state from $X_0^i$ to $X_t^i$ ($\forall i$). Define $\tilde{\mathbf{X}}_t$ as a set of auxiliary variables such that*

$$q(\tilde{\boldsymbol{x}}_t|\boldsymbol{x}_{t+1}) = p(\mathbf{X}_0^I = \tilde{\boldsymbol{x}}_t^I|\mathbf{X}_0^J = \boldsymbol{x}_{t+1}^J) \cdot \mathbb{1}[\tilde{\boldsymbol{x}}_t^J = \boldsymbol{x}_{t+1}^J]. \tag{5}$$

*Then, the distribution $q(\mathbf{X}_t|\tilde{\boldsymbol{x}}_t, \boldsymbol{x}_{t+1})$ is the following: $q(\mathbf{X}_t|\tilde{\boldsymbol{x}}_t, \boldsymbol{x}_{t+1}) = \prod_i q(x_t^i|\tilde{x}_t^i, x_{t+1}^i)$.*

*– For $i \in I$, $q(x_t^i|\tilde{x}_t^i, x_{t+1}^i)$ equals $\alpha_t/\alpha_{t+1}$ if $x_t^i = $ <MASK> and equals $1 - \alpha_t/\alpha_{t+1}$ if $x_t^i = \tilde{x}_t^i$.*

*– For $i \in J$, $q(x_t^i|\tilde{x}_t^i, x_{t+1}^i) = 1$ if and only if $x_t^i = x_{t+1}^i$.*

Since $q(\mathbf{X}_t|\tilde{\boldsymbol{x}}_t, \boldsymbol{x}_{t+1})$ is fully factorized, the copula model only needs to account for inter-variable dependencies in $q(\tilde{\mathbf{X}}_t|\boldsymbol{x}_{t+1})$. Following Equation (5), we can transform $p_{\mathrm{copula}}(\mathbf{X}_0)$, which estimates the clean data distribution, into $p_{\mathrm{copula}}(\tilde{\mathbf{X}}_t|\boldsymbol{x}_{t+1})$ that approximates $q(\tilde{\mathbf{X}}_t|\boldsymbol{x}_{t+1})$ by conditioning it on the unmasked tokens in $\boldsymbol{x}_{t+1}$ (i.e., $\boldsymbol{x}_{t+1}^J$). Specifically, for autoregressive copula models (i.e., $p_{\mathrm{copula}}(\boldsymbol{x}) := \prod_i p_{\mathrm{copula}}(x_i|\boldsymbol{x}_{<i})$), we construct $p_{\mathrm{copula}}(\tilde{\mathbf{X}}_t|\boldsymbol{x}_{t+1})$ by conditioning each variable on the corresponding preceding tokens in $\boldsymbol{x}_{t+1}^J$ while enforcing $\tilde{x}_t^j = x_{t+1}^j$ ($\forall j \in J$):

$$p_{\mathrm{copula}}(\tilde{\boldsymbol{x}}_t|\boldsymbol{x}_{t+1}) := \prod_{i \in I} p_{\mathrm{copula}}(X_0^i = \tilde{x}_t^i|\mathbf{X}_0^{<i} = \tilde{\boldsymbol{x}}_t^{<i}) \cdot \prod_{j \in J} \mathbb{1}[\tilde{x}_t^j = x_{t+1}^j]. \tag{6}$$

This copula distribution is biased even if the autoregressive model perfectly captures the data distribution since it cannot condition on subsequent unmasked tokens in $\boldsymbol{x}_{t+1}$. In contrast, while being able to condition on all unmasked tokens, diffusion models cannot capture dependence between variables. Combining the two estimates in a proper way will lead to better empirical performance.

Continuing with the example in Figure 2, we assume an autoregressive copula model encodes the probability table on the left. As shown on the right, when provided with the suffix prompt "in Switzerland", the copula model alone cannot adjust its probabilities, as it can only condition on prefix prompts. However, a diffusion model that captures the strong dependence between "Switzerland" and $Y = $ "skiing" can, through I-projection, set the correct marginal probabilities of $Y$, while keeping the copula unchanged. This allows the model to reliably generate "how about alpine skiing."

---

**Algorithm 1** Draw samples from a discrete diffusion model with the help of a copula model

---
1: **Inputs:** a diffusion model $p_{\mathrm{dm}}$, a copula model $p_{\mathrm{copula}}$, number of time steps $T$
2: **Outputs:** a sample $\boldsymbol{x}_0$ from the discrete diffusion model augmented by the copula model
3: **Initialize:** Sample $\boldsymbol{x}_T$ from the prior noise distribution $p(\mathbf{X}_T)$
4: **for** $t = T-1$ to $0$
5:    | Compute $\{p_{\mathrm{dm}}(\tilde{X}_t^i|\boldsymbol{x}_{t+1})\}_i$ and $\{p_{\mathrm{dm}}(\tilde{X}_t^i|\boldsymbol{x}_{t+1}^{<i})\}_i$ using the diffusion model
6:    | Compute $\mathbf{V}[i,\tilde{x}_t^i] = \log p_{\mathrm{dm}}(\tilde{x}_t^i|\boldsymbol{x}_{t+1}) - \log p_{\mathrm{dm}}(\tilde{x}_t^i|\boldsymbol{x}_{t+1}^{<i})$ $(\forall i, \tilde{x}_t^i)$ following Equation (10)
7:    | Sample $\tilde{\boldsymbol{x}}_t$ from $\hat{p}(\tilde{\boldsymbol{x}}_t|\boldsymbol{x}_{t+1}) \propto p_{\mathrm{copula}}(\tilde{\boldsymbol{x}}_t|\boldsymbol{x}_{t+1}) \cdot \prod_i \exp(\mathbf{V}[i,\tilde{x}_t^i])$ ($p_{\mathrm{copula}}$ is defined by Equation (6))
8:    | Sample $\boldsymbol{x}_t$ from $q(\mathbf{X}_t|\tilde{\boldsymbol{x}}_t, \boldsymbol{x}_{t+1})$ (defined in Proposition 5)

---

Lastly, we need the univariate marginals of $q(\tilde{\mathbf{X}}_t|\boldsymbol{x}_{t+1})$, which can be derived by renormalizing $\{q(X_t^i|\boldsymbol{x}_{t+1})\}_i$ to zero out the probability of the mask state according to the following result.

**Proposition 6.** *For each $i$ and data category $c \neq$ <MASK>, $q(\tilde{X}_t^i = c|\boldsymbol{x}_{t+1}) \propto q(X_t^i = c|\boldsymbol{x}_{t+1})$.*

As a result, for each $i$, the distribution $p_{\mathrm{dm}}(\tilde{X}_t^i|\boldsymbol{x}_{t+1})$ can be similarly obtained by renormalizing $p_{\mathrm{dm}}(X_t^i|\boldsymbol{x}_{t+1})$, which is directly obtained from the denoising model, to exclude the mask state.

## 5.2 Approximate I-Projection with Autoregressive Models

Given univariate marginals $\{p_{\mathrm{dm}}(\tilde{X}_t^i|\boldsymbol{x}_{t+1})\}_i$ and an autoregressive copula distribution $p_{\mathrm{copula}}(\tilde{\mathbf{X}}_t|\boldsymbol{x}_{t+1})$, both of which estimate the target distribution $q(\tilde{\mathbf{X}}_t|\boldsymbol{x}_{t+1})$, our goal is to combine them following the I-projection procedure described in Section 4.1. Specifically, this involves solving the convex optimization problem in Equation (4), which is specialized to the following:

$$\sum_{\tilde{\boldsymbol{x}}_t} p_{\mathrm{copula}}(\tilde{\boldsymbol{x}}_t|\boldsymbol{x}_{t+1}) \cdot \prod_i \exp(\mathbf{V}[i,\tilde{x}_t^i]) - \sum_{i=1}^N \sum_{\tilde{x}_t=1}^C \mathbf{V}[i,\tilde{x}_t^i] \cdot p_{\mathrm{dm}}(\tilde{x}_t^i|\boldsymbol{x}_{t+1}). \tag{7}$$

Following Theorem 1, if $\mathbf{V}$ minimizes Equation (7), then the distribution defined by $\hat{p}(\tilde{\boldsymbol{x}}_t|\boldsymbol{x}_{t+1}) = p_{\mathrm{copula}}(\tilde{\boldsymbol{x}}_t|\boldsymbol{x}_{t+1}) \cdot \prod_i \exp(\mathbf{V}[i,\tilde{x}_t^i])$ is the I-projection of $p_{\mathrm{copula}}(\tilde{\boldsymbol{x}}_t|\boldsymbol{x}_{t+1})$ onto the set of distributions with the univariate marginals $\{p_{\mathrm{dm}}(\tilde{X}_t^i|\boldsymbol{x}_{t+1})\}_i$, which is the desired combined distribution.

Consider initializing all coefficients in $\mathbf{V}$ to zero, i.e., $\hat{p}(\tilde{\boldsymbol{x}}_t|\boldsymbol{x}_{t+1}) = p_{\mathrm{copula}}(\tilde{\boldsymbol{x}}_t|\boldsymbol{x}_{t+1})$. For each row $i$, if we only optimize the values $\mathbf{V}[i,:]$ and fix the rest to zero, the optimal coefficients are

$$\forall c, \ \mathbf{V}[i,c] = \log p_{\mathrm{dm}}(\tilde{X}_t^i = c|\boldsymbol{x}_{t+1}) - \log p_{\mathrm{copula}}(\tilde{X}_t^i = c|\boldsymbol{x}_{t+1}). \tag{8}$$

We approximate the solution to Equation (7) by applying the above update (Eq. (8)) to each row in $\mathbf{V}$ independently, as it strikes a proper balance between efficiency and empirical performance.

While the first term on the right-hand side of Equation (8) can be acquired from the diffusion model, the second term is not accessible through the copula model. Plug in the definition in Equation (6), the required marginal probabilities can be written as (for $j \in J$, $p_{\mathrm{copula}}(\tilde{x}_t^j|\boldsymbol{x}_{t+1}) = 1$ iff $\tilde{x}_t^j = x_{t+1}^j$)

$$\forall i \in I, \ p_{\mathrm{copula}}(\tilde{x}_t^i|\boldsymbol{x}_{t+1}) = p_{\mathrm{copula}}(X_i = \tilde{x}_t^i|\mathbf{X}_{K_i} = \boldsymbol{x}_{t+1}^{K_i}), \ \text{where } K_i = \{j : j \in J \text{ and } j < i\}. \tag{9}$$

The above probabilities cannot be computed from the autoregressive model since we need to "marginalize out" preceding tokens that are not in $K_i$ (i.e., those not given as evidence in $\boldsymbol{x}_{t+1}$). However, these terms can be estimated using the diffusion model. Assume both the diffusion model and the autoregressive model perfectly encode the data distribution. According to Proposition 6, the diffusion model computes $p_{\mathrm{dm}}(\tilde{X}_t^i|\boldsymbol{x}_{t+1}) = q(\tilde{X}_t^i|\boldsymbol{x}_{t+1})$. Comparing it to Equation (9), which gives $p_{\mathrm{copula}}(\tilde{X}_t^i|\boldsymbol{x}_{t+1}) = q(\tilde{X}_t^i|\boldsymbol{x}_{t+1}^{K_i})$, we only need to additionally restrict the diffusion model to only condition on preceding unmasked tokens in $\boldsymbol{x}_{t+1}$, since $K_i$ is the intersection of $J$ and $\{j : j < i\}$. Therefore, if both models well-approximate the data distribution, we have $p_{\mathrm{copula}}(\tilde{x}_t^i|\boldsymbol{x}_{t+1}) \approx q(\tilde{x}_t^i|\boldsymbol{x}_{t+1}^{K_i}) = q(\tilde{x}_t^i|\boldsymbol{x}_{t+1}^{<i}) \approx p_{\mathrm{dm}}(\tilde{x}_t^i|\boldsymbol{x}_{t+1}^{<i})$, where the equality holds since all values in $\boldsymbol{x}_{t+1}^{<i}$ but not in $\boldsymbol{x}_{t+1}^{K_i}$ are <MASK>, and does not "contribute to" the distribution of $\tilde{X}_t^i$ according to Proposition 5). Correspondingly, we update $\mathbf{V}$ following

$$\forall i, c, \ \mathbf{V}[i,c] = \log p_{\mathrm{dm}}(\tilde{X}_t^i = c|\boldsymbol{x}_{t+1}) - \log p_{\mathrm{dm}}(\tilde{X}_t^i = c|\boldsymbol{x}_{t+1}^{<i}). \tag{10}$$

For denoising neural networks that are implemented with bidirectional Transformers, we can simply apply causal attention masks to the self-attention layers to obtain $\{p_{\mathrm{dm}}(\tilde{X}_t^i|\boldsymbol{x}_{t+1}^{<i})\}_i$.

## 5.3 THE OVERALL DIFFUSION SAMPLING PROCESS

Given a diffusion model $p_{\mathrm{dm}}$ and an autoregressive copula model $p_{\mathrm{copula}}$, the sampling procedure is outlined in Algorithm 1. First, we sample $\boldsymbol{x}_T$ from the prior noise distribution $p(\mathbf{X}_T)$ (line 3). During each denoising step $t$, we compute the univariate marginals $\{p_{\mathrm{dm}}(\tilde{X}_t^i|\boldsymbol{x}_{t+1})\}_i$ and $\{p_{\mathrm{dm}}(\tilde{X}_t^i|\boldsymbol{x}_{t+1}^{<i})\}_i$ based on the previously obtained $\boldsymbol{x}_{t+1}$ (line 5). These marginals are then used to compute the entries in $\mathbf{V}$ (line 6), which approximates the I-projection of $p_{\mathrm{copula}}(\tilde{\mathbf{X}}_t|\boldsymbol{x}_{t+1})$ onto the set of distributions with univariate marginals $\{p_{\mathrm{dm}}(\tilde{X}_t^i|\boldsymbol{x}_{t+1})\}_i$ (cf. Sec. 5.2).

Afterwards, we sample $\tilde{\boldsymbol{x}}_t$ from the combined distribution $\hat{p}(\tilde{\mathbf{X}}_t|\boldsymbol{x}_{t+1})$ (line 7). Specifically, following Equation (6), we sample autoregressively following $\hat{p}(\tilde{\boldsymbol{x}}_t|\boldsymbol{x}_{t+1}) = \prod_i \hat{p}(\tilde{x}_t^i|\boldsymbol{x}_{t+1}, \tilde{\boldsymbol{x}}_t^{<i})$, where

$$\hat{p}(\tilde{x}_t^i|\boldsymbol{x}_{t+1}, \tilde{\boldsymbol{x}}_t^{<i}) \propto p_{\mathrm{copula}}(X_i = \tilde{x}_t^i|\mathbf{X}_{<t} = \tilde{\boldsymbol{x}}_t^{<i}) \cdot \exp(\mathbf{V}[i, \tilde{x}_t^i]) \cdot \mathbb{1}[\tilde{x}_t^i = x_{t+1}^i].$$

Finally, we sample $\boldsymbol{x}_t$ from $q(\mathbf{X}_t|\tilde{\boldsymbol{x}}_t, \boldsymbol{x}_{t+1})$ (line 8) as defined in Proposition 5. To improve the algorithm's efficiency, we introduce a variant that unmasks tokens in an autoregressive manner. Specifically, at step $t$, all tokens except the first $(T-t)/T$ portion of the tokens in $\boldsymbol{x}_t$ are converted to `<MASK>`. Since $\hat{p}$ is sampled autoregressively, this allows us to use techniques such as KV-caching for autoregressive Transformers (Pope et al., 2023) to significantly reduce computation cost introduced by the copula model. See Appendix E for details and the concrete algorithm.

## 6 EXPERIMENTS

We empirically validate the proposed method, **D**iscrete **C**opula **D**iffusion (DCD), on language modeling tasks (Sec. 6.1 and 6.2) and antibody sequence infilling tasks (Sec. 6.3). For all tasks, we evaluate whether DCD can effectively reduce the number of diffusion steps while maintaining strong performance. Specifically, since DCD combines two pretrained models: a discrete diffusion model and an autoregressive copula model, we examine whether DCD outperforms each individual model.

## 6.1 UNCONDITIONAL TEXT GENERATION

We first compare the quality of unconditional samples generated by models trained on either Web-Text (Radford et al., 2019) or OpenWebText (Gokaslan & Cohen, 2019), which contain web content extracted from URLs shared on Reddit with a minimum number of upvotes. We adopt the medium-sized SEDD model (Lou et al., 2024) (SEDD$_\mathrm{M}$) since it is a SoTA discrete diffusion model for text generation. The GPT-2-small model (Radford et al., 2019) (GPT-2$_\mathrm{S}$) serves as the copula model.

We generate samples of 128 tokens each. Following Han et al. (2023); Dieleman et al. (2022), we evaluate sample quality using their generative perplexity, which is the perplexity of the samples when evaluated with the GPT-2-large model. Since previous studies have observed that this metric can be affected by distribution annealing methods such as nucleus sampling, we always sample directly from the models. SEDD$_\mathrm{M}$ is evaluated with 2 to 256 diffusion steps and DCD (i.e., SEDD$_\mathrm{M}$ with GPT-2$_\mathrm{S}$ as the copula model) is run with diffusion steps ranging from 2 to 32. We adopt the log-linear noise schedule suggested by the SEDD paper. See Appendix G.1 for more details.

For each configuration, we draw 10,000 samples and report the average perplexity in Figure 3. First, when fixing the number of denoising steps between 2 to 32, we observe that DCD outperforms both SEDD$_\mathrm{M}$ with the same number of denoising steps and GPT-2$_\mathrm{S}$. This provides empirical validation of the effectiveness of the I-projection procedure for modeling inter-variable dependencies.

Additionally, DCD with just 4 denoising steps achieves performance comparable to SEDD$_\mathrm{M}$ with 128 steps, representing a 32x reduction in the number of denoising steps. This result not only demonstrates the efficiency of DCD but also underscores the importance of modeling inter-variable dependencies in discrete diffusion models, particularly in few-step generation settings.

Finally, as shown in Figure 4, SEDD fails to generate fluent and meaningful sentences given only a few diffusion steps, as too many tokens have to be generated in each step. In contrast, by modeling the inter-variable dependencies, DCD generates smooth sentences with only 4 denoising steps.

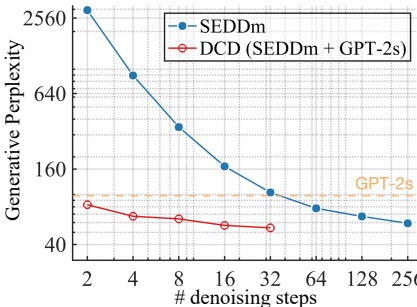

Figure 3: Generative perplexity (↓) with different numbers of denoising steps.

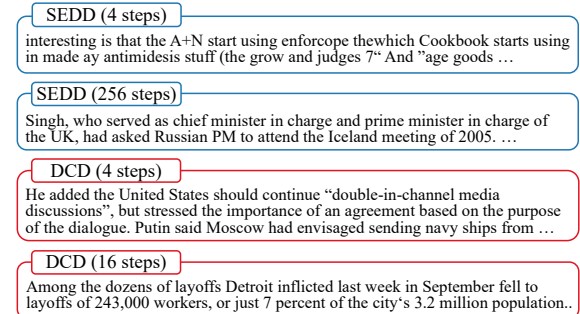

Figure 4: Generated text from SEDD_M and DCD with different number of steps. See Appendix I for more.

Table 1: Evaluation of text infilling performance using the MAUVE score (↑) with 5 prompt masks. Scores of DCD are all better than (i) SEDD with the same # denoising steps, and (ii) GPT-2_S.

| Prompt ranges (remainder is masked) | SSD-LM | | GPT-2_S | SEDD_M | | | | | DCD (ours) | | | | |
|---|---|---|---|---|---|---|---|---|---|---|---|---|---|
| | 100 | 500 | N/A | 2 | 4 | 8 | 16 | 32 | 2 | 4 | 8 | 16 | 32 |
| [0.1,0.2] & [0.5,0.7] | 0.057 | 0.083 | 0.079 | 0.013 | 0.051 | 0.122 | 0.152 | 0.201 | 0.158 | 0.187 | 0.185 | 0.195 | 0.211 |
| [0.25,0.75] | 0.072 | 0.108 | 0.188 | 0.027 | 0.110 | 0.226 | 0.237 | 0.278 | 0.249 | 0.251 | 0.257 | 0.314 | 0.298 |
| [0.0,0.1] & [0.4,0.6] & [0.9,1.0] | 0.333 | 0.681 | 0.928 | 0.827 | 0.940 | 0.972 | 0.980 | 0.979 | 0.962 | 0.976 | 0.979 | 0.982 | 0.983 |
| [0.4,0.5] & [0.8,1.0] | 0.436 | 0.565 | 0.914 | 0.896 | 0.944 | 0.978 | 0.978 | 0.980 | 0.963 | 0.975 | 0.975 | 0.976 | 0.981 |
| [0.2,0.3] & [0.6,0.8] | 0.041 | 0.054 | 0.069 | 0.016 | 0.056 | 0.128 | 0.207 | 0.215 | 0.171 | 0.178 | 0.215 | 0.217 | 0.403 |

**Efficiency.** We compare the sample time and the generative perplexity of DCD against competitive baselines in Figure 5. We additionally adopt another recent discrete diffusion baseline MDLM (Sahoo et al., 2024). We adopt the autoregressive version of DCD as described in Section 5.3 and Appendix E. Compared to the baselines, DCD consistently achieves better generative perplexity given a fixed runtime constraint. It also requires less time to reach a desired perplexity value. We defer a comprehensive study of DCD's efficiency to Appendix F.

## 6.2 CONDITIONAL TEXT GENERATION

We now move on to conditional text generation, where certain tokens are provided in advance, and the task is to generate the remaining tokens. As shown in the first column of Table 1, we use five mask strategies, where tokens in specific prompt ranges are given (we use a sequence length of 128). We adopt the MAUVE score (Pillutla et al., 2021) with the default settings to compare the difference between the generated and original texts. See Appendix G.2 for further details.

For all methods, we use the same set of 2,000 text sequences from the validation set of WikiText-103 (Merity et al., 2022). After applying the prompt mask, we generate 5 samples for each prompt, resulting in a total number of 10,000 samples.

In addition to SEDD_M and GPT-2_S, we compare against SSD-LM (Han et al., 2023), which is a semi-autoregressive diffusion model designed for text infilling. We adopt the autoregressive unmasking variant of DCD described in the last paragraph of Section 5.3.

Results are presented in Table 1. First, DCD outperforms all three baselines in all five tasks. Additionally, when fixing the number of denoising steps between 2 and 32, DCD surpasses both of its base models. Notably, while both GPT-2_S and the 2-step SEDD_M performs poorly on the first, the second, and the fifth tasks, combining them in a principled way allows DCD to achieve significantly better performance using only two denoising steps.

## 6.3 ANTIBODY SEQUENCE INFILLING

We consider the task of unguided antibody infilling, where certain complementarity determining regions (CDRs) of antibodies (i.e., sequences of amino acids) are missing and to be generated by the model. We adopt NOC-D (Gruver et al., 2023), which is a discrete diffusion model trained on 104K

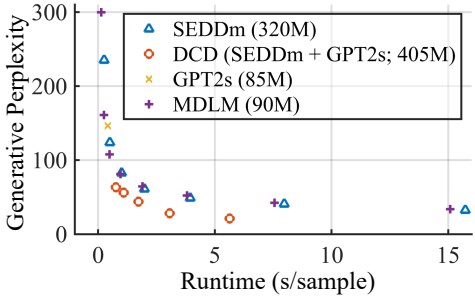

Figure 5: Sampling time vs. generative perplexity (the autoregressive version of DCD is used).

Figure 6: Antibody sequence infilling performance measured by sequence recovery rate (↑). We compare DCD against its two base models in two tasks, where amino acids at different locations are masked. DCD outperforms both baselines with only 4 denoising steps.

| Method | # steps | Task | |
|--------|---------|------|------|
| | | HCDR{1+2+3} | {H+L}CDR1 |
| GPT | N/A | 57.21 | 90.28 |
| NOS-D | 64 | 51.56 | 88.82 |
| DCD | 4 | **58.28** | **91.58** |

antibody sequences from the Observed Antibody Space dataset (Ruffolo et al., 2023). We further train a GPT model on the same dataset as the copula model. See Appendix G.3 for training details.

We follow Gruver et al. (2023) to select the same 10 antibody seed sequences from paired OAS (Olsen et al., 2022). We consider two infilling tasks: (i) three CDRs {HCDR1, HCDR2, HCDR3} are masked, and (ii) two CDRs {HCDR1, LCDR1} are masked. We follow the original paper and run 64 diffusion steps for NOS-D. For DCD (i.e., combining NOS-D with the trained GPT model as the copula model), we use 4 denoising steps. We measure the sequence recovery rate, i.e., the accuracy of the infilled sequences given the ground truth sequence.

As shown in Figure 6, by combining the univariate marginals from NOS-D and the dependencies captured by the GPT model, DCD can also perform well in antibody sequence infilling tasks.

## 7 RELATED WORK AND CONCLUSION

Diffusion models have been widely applied to model discrete data such as text and DNA sequences. Encouraged by the successes of continuous diffusion models (e.g., Ho et al. (2020); Song et al. (2020)), initial attempts convert discrete data into continuous embeddings with either predefined or learned mappings. This enables the use of continuous diffusion models for discrete data (Chen et al., 2022; Dieleman et al., 2022; Li et al., 2022; Lovelace et al., 2023). However, due to the need for ad-hoc mappings between the discrete data space and the continuous embedding space, which have to be pre-defined or pre-trained, continuous diffusion models are not as effective for modeling discrete distributions (Strudel et al., 2022; Li et al., 2022; Dieleman et al., 2022).

Austin et al. (2021) proposed the first diffusion model designed directly for discrete data. Later works further improved discrete diffusion models from various aspects such as better loss functions/learning objectives (Campbell et al., 2022; Meng et al., 2022; Lou et al., 2024; Benton et al., 2022), better model architectures (Sun et al., 2022), better sampling algorithms (Chen et al., 2023), and unifying and scaling up existing techniques (Sahoo et al., 2024; Shi et al., 2024).

Despite the recent breakthroughs of discrete diffusion models, few papers address the challenge of sampling in a few denoising steps. Some works attribute the failure to perform high-quality few-step generation to a scaling problem of the model. However, we show that the fundamental problem lies in the assumption made by discrete diffusion models that each variable is denoised independently at each step. In addition to identifying this problem, we propose a general solution Discrete Copula Diffusion that combines a discrete diffusion model with a copula model at inference time to obtain a better estimate of the denoising distribution at each step. Concurrently, Guo et al. (2024) show that energy-based models can also be used as copula models to capture inter-variable dependencies.

There are a few limitations of DCD. First, in addition to a discrete diffusion model, it requires another copula model, which may require additional training for certain applications. Second, although the I-projected distribution is guaranteed as a better estimate of the target distribution, the I-projection step often needs to be approximated in practice. Finally, although DCD requires fewer denoising steps, the computation cost of each step is higher than in discrete diffusion models. Therefore, DCD may not always provide a notable speedup. However, DCD points out the inter-dependency modeling problem and opens up the possibility of combining different types of generative models for better overall performance.

ACKNOWLEDGEMENTS

This work was funded in part by the DARPA ANSR program under award FA8750-23-2-0004, the DARPA CODORD program under award HR00112590089, the Deutsche Forschungsgemeinschaft (DFG, German Research Foundation) under Germany's Excellence Strategy - EXC 2075 – 390740016, NSF grant #IIS-1943641, and gifts from Adobe Research and Amazon. We acknowledge the support of the Stuttgart Center for Simulation Science (SimTech). MN thanks IMPRS-IS (International Max Planck Research School for Intelligent Systems) for the support.

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

## A    PROOF OF THE THEORETICAL RESULTS

*Proof of Proposition 1.* Following (Ho et al., 2020; Sohl-Dickstein et al., 2015), the negative ELBO $\mathcal{L}$ can be decomposed as follows:

$$
\begin{aligned}
\mathcal{L} &= \mathbb{E}_q \left[ -\log p(\boldsymbol{x}_T) - \sum_{t=1}^{T} \log \frac{p_\theta(\boldsymbol{x}_{t-1}|\boldsymbol{x}_t)}{q(\boldsymbol{x}_t|\boldsymbol{x}_{t-1})} \right], \\
&= \mathbb{E}_q \left[ -\log p(\boldsymbol{x}_T) - \sum_{t=1}^{T} \log \frac{p_\theta(\boldsymbol{x}_{t-1}|\boldsymbol{x}_t) \cdot q(\boldsymbol{x}_{t-1})}{q(\boldsymbol{x}_{t-1}|\boldsymbol{x}_t) \cdot q(\boldsymbol{x}_t)} \right], \\
&= \mathbb{E}_q \left[ -\log \frac{p(\boldsymbol{x}_T)}{q(\boldsymbol{x}_T)} - \sum_{t=1}^{T} \log \frac{p_\theta(\boldsymbol{x}_{t-1}|\boldsymbol{x}_t)}{q(\boldsymbol{x}_{t-1}|\boldsymbol{x}_t)} - \log p(\boldsymbol{x}_0) \right], \\
&= \mathrm{D}_{\mathrm{KL}}(q(\boldsymbol{x}_T) \parallel p(\boldsymbol{x}_T)) + \mathbb{E}_q \left[ \sum_{t=1}^{T} \mathrm{D}_{\mathrm{KL}}(q(\boldsymbol{x}_{t-1}|\boldsymbol{x}_t) \parallel p_\theta(\boldsymbol{x}_{t-1}|\boldsymbol{x}_t)) \right] + \mathrm{H}(\boldsymbol{x}_0). \quad (11)
\end{aligned}
$$

The first term equals 0 as we assume the noise distribution $p(\mathbf{X}_T)$ is consistent in the noising and the denoising processes. Given the independent denoising assumption, when the denoising distribution are optimal, we have

$$
\forall t \in \{1, \dots, T\}, \ p_\theta(\boldsymbol{x}_{t-1}|\boldsymbol{x}_t) = \prod_i q(x_{t-1}^i|\boldsymbol{x}_t).
$$

Plug in Equation (11) and using the definition of total correlation, we have:

$$
\begin{aligned}
\mathcal{L} &= \mathrm{D}_{\mathrm{KL}}(q(\boldsymbol{x}_T) \parallel p(\boldsymbol{x}_T)) + \mathbb{E}_q \left[ \sum_{t=1}^{T} \mathrm{D}_{\mathrm{KL}}(q(\boldsymbol{x}_{t-1}|\boldsymbol{x}_t) \parallel \prod_i q(x_{t-1}^i|\boldsymbol{x}_t)) \right] + \mathrm{H}(\boldsymbol{x}_0) \\
&= \mathrm{D}_{\mathrm{KL}}(q(\boldsymbol{x}_T) \parallel p(\boldsymbol{x}_T)) + \sum_{t=1}^{T} \mathrm{D}_{\mathrm{TC}}(q(\mathbf{X}_{t-1}|\mathbf{X}_t)) + \mathrm{H}(p(\mathbf{X}_0)) \\
&\geq \mathrm{H}(p(\mathbf{X}_0)) + \sum_{t=1}^{T} \mathrm{D}_{\mathrm{TC}}(q(\mathbf{X}_{t-1}|\mathbf{X}_t)).
\end{aligned}
$$

$\square$

*Proof of Proposition 2.* According to Pythagoras' triangle-inequality theorem, if $\hat{p}$ is the I-projection of $p_{\mathrm{est}}$ onto $\mathcal{P}_{\mathrm{mar}}^{p_{\mathrm{tar}}}$, and $\mathcal{P}_{\mathrm{mar}}^{p_{\mathrm{tar}}}$ is convex (this can be shown by applying the definition of a convex set), the following holds for any $p' \in \mathcal{P}_{\mathrm{mar}}^{p_{\mathrm{tar}}}$:

$$
\mathrm{D}_{\mathrm{KL}}(p' \parallel p_{\mathrm{est}}) \geq \mathrm{D}_{\mathrm{KL}}(p' \parallel \hat{p}) + \mathrm{D}_{\mathrm{KL}}(\hat{p} \parallel p_{\mathrm{est}}). \quad (12)
$$

Choosing $p' = p_{\mathrm{tar}}$, we have

$$
\mathrm{D}_{\mathrm{KL}}(p_{\mathrm{tar}} \parallel \hat{p}) \leq \mathrm{D}_{\mathrm{KL}}(p_{\mathrm{tar}} \parallel p_{\mathrm{est}}) - \mathrm{D}_{\mathrm{KL}}(\hat{p} \parallel p_{\mathrm{est}}) < \mathrm{D}_{\mathrm{KL}}(p_{\mathrm{tar}} \parallel p_{\mathrm{est}}),
$$

where the last inequality holds since $\mathrm{D}_{\mathrm{KL}}(\hat{p} \parallel p_{\mathrm{est}}) > 0$ if the set of univariate marginals of $p_{\mathrm{est}}$ and $p_{\mathrm{tar}}$ are different (as assumed in the proposition).

$\square$

*Proof of Proposition 3.* Following the definition of $\hat{p}$, we write down the constrained optimization problem as follows

$$
\underset{p'}{\mathrm{minimize}} \ \mathrm{D}_{\mathrm{KL}}(p' \parallel p_{\mathrm{est}})
$$

$$
\mathrm{s.t.} \ \forall i \in \{1, \dots, N\}, x_i \in \{1, \dots, C\}, \ \sum_{\boldsymbol{x}_{\backslash i}} p'(\boldsymbol{x}_{\backslash i}, x_i) = p_{\mathrm{tar}}(x_i).
$$

To incorporate the constraints, we use the method of Lagrange multipliers. The Lagrangian for this problem is

$$\mathcal{L}(p', \{\lambda_i\}_{i=1}^N) = \sum_{\boldsymbol{x}} p'(\boldsymbol{x}) \log \frac{p'(\boldsymbol{x})}{p_{\text{est}}(\boldsymbol{x})} + \sum_{i=1}^N \sum_{x_i=1}^C \lambda_i(x_i) \cdot \left( \sum_{\boldsymbol{x}_{\backslash i}} p'(\boldsymbol{x}_{\backslash i}, x_i) - p_{\text{tar}}(x_i) \right),$$

where the Lagrange multipliers $\{\lambda_i\}_{i=1}^N$ enforce the univariate marginal constraints.

To minimize the Lagrangian with respect to $p'(\boldsymbol{x})$, we take the partial derivative of $\mathcal{L}(p', \{\lambda_i\}_{i=1}^N)$ with respect to $p'(\boldsymbol{x})$ and set it to 0:

$$\frac{\partial \mathcal{L}(p', \{\lambda_i\}_{i=1}^N)}{\partial p'(\boldsymbol{x})} = \log \frac{p'(\boldsymbol{x})}{p_{\text{est}}(\boldsymbol{x})} + 1 + \sum_i \lambda_i(x_i) = 0.$$

Simplifying this equation gives

$$p'(\boldsymbol{x}) = p_{\text{est}}(\boldsymbol{x}) \cdot \exp\left( -1 - \sum_i \lambda_i(x_i) \right).$$

Defining $\sigma_i(x_i) := \exp(-\lambda_i(x_i) - 1/N)$ gives $p'(\boldsymbol{x}) = p_{\text{est}}(\boldsymbol{x}) \prod_i \sigma_i(x_i)$.

Existence of the solution follows from the fact that (i) the objective function is convex and bounded (since probability values are in $[0, 1]$), and (ii) the set of constraints is feasible (e.g., $p'(\boldsymbol{x}) = \prod_i p_{\text{tar}}(x_i)$ or $p'(\boldsymbol{x}) = p_{\text{tar}}(\boldsymbol{x})$).

$\square$

*Proof of Theorem 1.* We show that for any $\mathbf{V}^*$ that minimizes the objective function $\mathcal{L}(\mathbf{V}; p_{\text{tar}}, p_{\text{est}})$, the corresponding $p'$ defined by $p'(\boldsymbol{x}) = p_{\text{est}}(\boldsymbol{x}) \cdot \prod_i \exp(\mathbf{V}[i, x_i])$ belongs to the set $\mathcal{P}_{\text{mar}}^{p_{\text{tar}}}$. Specifically, for any $\mathbf{V}$ that minimizes the objective, the partial derivative of $\mathcal{L}(\mathbf{V}; p_{\text{tar}}, p_{\text{est}})$ with respect to any $\mathbf{V}[i, x_i]$ should be 0:

$$\frac{\partial \mathcal{L}(\mathbf{V}; p_{\text{tar}}, p_{\text{est}})}{\partial \mathbf{V}[i, x_i]} = \exp(\mathbf{V}[i, x_i]) \sum_{\boldsymbol{x}_{\backslash i}} p_{\text{est}}(\boldsymbol{x}_{\backslash i}, x_i) \prod_{j \neq i} \exp(\mathbf{V}[j, x_j]) - p_{\text{tar}}(x_i) = 0.$$

Plug in the definition of $p'$, we have

$$0 = \sum_{\boldsymbol{x}_{\backslash i}} p'(\boldsymbol{x}_{\backslash i}, x_i) - p_{\text{tar}}(x_i) = p'(x_i) - p_{\text{tar}}(x_i). \tag{13}$$

Since Equation (13) holds for all $(i, x_i)$ pairs, we have that every minimizer of $\mathcal{L}(\mathbf{V}; p_{\text{tar}}, p_{\text{est}})$ corresponds to a distribution $p'$ in $\mathcal{P}_{\text{mar}}^{p_{\text{tar}}}$. Since $\mathcal{L}(\mathbf{V}; p_{\text{tar}}, p_{\text{est}})$ is convex, we can also argue the converse: if a distribution $p'$ with the above-defined form belongs to $\mathcal{P}_{\text{mar}}^{p_{\text{tar}}}$, then the corresponding $\mathbf{V}$ is a minimizer of $\mathcal{L}(\mathbf{V}; p_{\text{tar}}, p_{\text{est}})$.

According to Proposition 3, the solution to the following I-projection exists and its solution $\hat{p}$ has the same form as $p'$.

$$\hat{p} = \arg\min_{p' \in \mathcal{P}_{\text{mar}}^p} \mathrm{D}_{\text{KL}}(p' \parallel p_{\text{est}}).$$

Since $\hat{p}$ has the same form as $p'$ (by Prop. 3) and belongs to $\mathcal{P}_{\text{mar}}^{p_{\text{tar}}}$, it is the a minimizer of $\mathcal{L}(\mathbf{V}; p_{\text{tar}}, p_{\text{est}})$.

$\square$

*Proof of Proposition 4.* The copula of $p$ is shown to be invariant under rescalings of the form $q(\boldsymbol{x}) \propto p(\boldsymbol{x}) \cdot \prod_i \exp(\mathbf{V}[i, x_i])$ for any $\mathbf{V} \in \mathbb{R}^{N \times C}$ by using the parameterization of a discrete copula by conditional odds ratios (Definition 2). The scaling factors cancel in the ratios as shown, e.g. by Rudas (2018, Theorem 12.3).

$\square$

*Proof of Proposition 5.* We start by writing the probability $q(\boldsymbol{x}_t|\boldsymbol{x}_{t+1})$ using the Bayes' rule:

$$q(\boldsymbol{x}_t|\boldsymbol{x}_{t+1}) = q(\boldsymbol{x}_{t+1}|\boldsymbol{x}_t) \cdot \frac{q(\boldsymbol{x}_t)}{q(\boldsymbol{x}_{t+1})},$$

$$= \sum_{\boldsymbol{x}_0} \frac{1}{q(\boldsymbol{x}_{t+1})} \cdot q(\boldsymbol{x}_{t+1}|\boldsymbol{x}_t) \cdot q(\boldsymbol{x}_t|\boldsymbol{x}_0) \cdot p(\boldsymbol{x}_0), \tag{14}$$

where the last equality follows from $q(\boldsymbol{x}_t) = \sum_{\boldsymbol{x}_0} q(\boldsymbol{x}_t|\boldsymbol{x}_0) \cdot p(\boldsymbol{x}_0)$. Recall from the proposition that $I$ is defined as the set of variables $i$ such that $x_{t+1}^i = \texttt{<MASK>}$ and $J$ is the complement of $I$.

First, we must have $x_t^j = x_{t+1}^j$ for $j \in J$ since for any other value of $X_t^j$, we have $q(\boldsymbol{x}_{t+1}|\boldsymbol{x}_t) = 0$ in Equation (14). As a result, $q(\boldsymbol{x}_t|\boldsymbol{x}_{t+1})$ is also zero.

We then move our attention to the variables in $I$. We first consider the probability $q(X_t^i = \texttt{<MASK>}|\boldsymbol{x}_{t+1})$ for any $i \in I$. Following Equation (14), we have

$$q(X_t^i = \texttt{<MASK>}|\boldsymbol{x}_{t+1}) = \sum_{\boldsymbol{x}_0} \sum_{\boldsymbol{x}_t^{\backslash i}} \frac{1}{q(\boldsymbol{x}_{t+1})} \cdot q(\boldsymbol{x}_{t+1}|\boldsymbol{x}_t) \cdot q(\boldsymbol{x}_t|\boldsymbol{x}_0) \cdot p(\boldsymbol{x}_0),$$

$$= \sum_{\boldsymbol{x}_t^{\backslash i}} \frac{1}{q(\boldsymbol{x}_{t+1})} \cdot q(\boldsymbol{x}_{t+1}|\boldsymbol{x}_t) \cdot q(\boldsymbol{x}_t),$$

$$= \frac{q(X_{t+1}^i = \texttt{<MASK>}|X_t^i = \texttt{<MASK>}) \cdot q(X_t^i = \texttt{<MASK>})}{q(X_{t+1}^i = \texttt{<MASK>})},$$

$$= \frac{q(X_t^i = \texttt{<MASK>})}{q(X_{t+1}^i = \texttt{<MASK>})} = \frac{\alpha_t}{\alpha_{t+1}}. \tag{15}$$

We then focus on $\mathbf{X}_t^I = \boldsymbol{x}_t^I$, where none of the value in $\boldsymbol{x}_t^I$ is $\texttt{<MASK>}$. Note that we also need to have $\mathbf{X}_t^J = \boldsymbol{x}_{t+1}^J$.

$$q(\boldsymbol{x}_t|\boldsymbol{x}_{t+1}) \propto \sum_{\boldsymbol{x}_0} q(\boldsymbol{x}_{t+1}|\boldsymbol{x}_t) \cdot q(\boldsymbol{x}_t|\boldsymbol{x}_0) \cdot q(\boldsymbol{x}_0),$$

$$\overset{(a)}{=} q(\boldsymbol{x}_{t+1}|\boldsymbol{x}_t) \cdot q(\mathbf{X}_0 = \boldsymbol{x}_t),$$

$$= \left(\frac{\alpha_{t+1} - \alpha_t}{1 - \alpha_t}\right)^{|I|} \cdot q(\mathbf{X}_0 = \boldsymbol{x}_t),$$

$$\propto q(\mathbf{X}_0 = \boldsymbol{x}_t), \tag{16}$$

where $p(\mathbf{X}_0)$ is the data distribution; $(a)$ follows from the fact that no value in $\boldsymbol{x}_t$ is $\texttt{<MASK>}$, hence $\boldsymbol{x}_0 = \boldsymbol{x}_t$; $\frac{\alpha_{t+1} - \alpha_t}{1 - \alpha_t}$ is the probability of transitioning into the mask state from time $t$ to time $t+1$.

Denote $\tilde{\mathbf{X}}_t$ as a set of variables with the same configuration and semantics as $\mathbf{X}_t$, with the only difference that the category $\texttt{<MASK>}$ is excluded. By following Equation (16) and apply normalization, we conclude that

$$q(\tilde{\boldsymbol{x}}_t|\boldsymbol{x}_{t+1}) = p(\mathbf{X}_0^I = \tilde{\boldsymbol{x}}_t^I|\mathbf{X}_0^J = \boldsymbol{x}_{t+1}^J) \cdot \mathbb{1}[\tilde{\boldsymbol{x}}_t^J = \boldsymbol{x}_{t+1}^J]. \tag{17}$$

This matches the definition in Equation (5).

Finally, we verify the correctness of the distribution $q(\mathbf{X}_t|\tilde{\boldsymbol{x}}_t, \boldsymbol{x}_{t+1})$ defined in the proposition by verifying the following for any $\boldsymbol{x}_t$

$$q(\boldsymbol{x}_t|\boldsymbol{x}_{t+1}) = \sum_{\tilde{\boldsymbol{x}}_t} q(\tilde{\boldsymbol{x}}_t|\boldsymbol{x}_{t+1}) \cdot q(\boldsymbol{x}_t|\tilde{\boldsymbol{x}}_t, \boldsymbol{x}_{t+1}). \tag{18}$$

Denote $K$ as the set of variables $i$ such that $\boldsymbol{x}_t = \texttt{<MASK>}$ and $L$ as its complement. First, if $L \subseteq J$ (i.e., $I \subseteq K$), then both the left-hand side (LHS) and the right-hand sides (RHS) are zero. Specifically, the RHS is zero since according to the definition, $\forall i \in J \, \& \, i \in K$, we have $q(x_t^i|\tilde{x}_t^i, x_{t+1}^i) = 0$.

Next, if $K \subseteq I$, we can decompose $q(\boldsymbol{x}_t|\boldsymbol{x}_{t+1})$ as follows

$$q(\boldsymbol{x}_t|\boldsymbol{x}_{t+1}) = q(\boldsymbol{x}_t^{I \backslash K}|\boldsymbol{x}_{t+1}) \cdot \prod_{i \in K} q(x_t^i|\boldsymbol{x}_{t+1}) \cdot \prod_{j \in J} q(x_t^j|\boldsymbol{x}_{t+1}). \tag{19}$$

For any $j \in J$, if $x_t^j \neq x_{t+1}^j$ then both the LHS and the RHS of Equation (18) are zero. Otherwise we always have $q(x_t^j | \boldsymbol{x}_{t+1}) = 1$. Therefore, Equation (19) can be further simplified as

$$q(\boldsymbol{x}_t | \boldsymbol{x}_{t+1}) = q(\boldsymbol{x}_t^{I \setminus K} | \boldsymbol{x}_{t+1}) \cdot \prod_{i \in K} q(x_t^i | \boldsymbol{x}_{t+1}). \tag{20}$$

We then proceed to simplify the RHS of Equation (18):

$$\sum_{\tilde{\boldsymbol{x}}_t} q(\tilde{\boldsymbol{x}}_t | \boldsymbol{x}_{t+1}) \cdot q(\boldsymbol{x}_t | \tilde{\boldsymbol{x}}_t, \boldsymbol{x}_{t+1}),$$

$$= \sum_{\tilde{\boldsymbol{x}}_t^K} q(\tilde{\boldsymbol{x}}_t^K, \tilde{\boldsymbol{x}}_t^{I \setminus K} | \boldsymbol{x}_{t+1}) \cdot \left( \frac{\alpha_t}{\alpha_{t+1}} \right)^{|K|} \cdot \left( \frac{\alpha_{t+1} - \alpha_t}{\alpha_{t+1}} \right)^{|I| - |K|},$$

$$\stackrel{(a)}{=} \sum_{\tilde{\boldsymbol{x}}_t^K} q(\tilde{\boldsymbol{x}}_t^K, \tilde{\boldsymbol{x}}_t^{I \setminus K} | \boldsymbol{x}_{t+1}) \cdot \left( \frac{\alpha_{t+1} - \alpha_t}{\alpha_{t+1}} \right)^{|I| - |K|} \cdot \prod_{i \in K} q(x_t^i | \boldsymbol{x}_{t+1}),$$

$$= q(\tilde{\boldsymbol{x}}_t^{I \setminus K} | \boldsymbol{x}_{t+1}) \cdot \left( \frac{\alpha_{t+1} - \alpha_t}{\alpha_{t+1}} \right)^{|I| - |K|} \cdot \prod_{i \in K} q(x_t^i | \boldsymbol{x}_{t+1}),$$

$$\stackrel{(b)}{\propto} p(\mathbf{X}_0^{I \setminus K} = \tilde{\boldsymbol{x}}_t^{I \setminus K}, \mathbf{X}_0^J = \tilde{\boldsymbol{x}}_t^J) \cdot \prod_{i \in K} q(x_t^i | \boldsymbol{x}_{t+1}),$$

$$\stackrel{(c)}{\propto} q(\mathbf{X}_t^{I \setminus K} = \tilde{\boldsymbol{x}}_t^{I \setminus K} | \boldsymbol{x}_{t+1}) \cdot \prod_{i \in K} q(x_t^i | \boldsymbol{x}_{t+1}), \tag{21}$$

where $(a)$ follows from Equation (15), $(b)$ applies the definition in Equation (17), and $(c)$ is a result of applying Equation (16) to the case where $\tilde{\boldsymbol{x}}_t^L = \{ \tilde{\boldsymbol{x}}_t^{I \setminus K}, \tilde{\boldsymbol{x}}_t^J \}$ are not <MASK>.

By combining Equations (21) and (20), we conclude that the LHS and the RHS of Equation (18) are proportional to each other. Since they are both properly-normalized distributions, they must also match exactly.

$\square$

*Proof of Proposition 6.* We first state a more detailed version of the proposition: for each variable $i$ and data category $c$ ($c \neq$ <MASK>), we have

$$q(\tilde{X}_t^i = c | \boldsymbol{x}_{t+1}) = \frac{1}{Z} \cdot q(X_t^i = c | \boldsymbol{x}_{t+1}), \text{ where } Z = \sum_{c \neq \text{<MASK>}} q(X_t^i = c | \boldsymbol{x}_{t+1}).$$

According to the proof of Proposition 5, Equation (18) holds for all $\boldsymbol{x}_t$. Therefore, we have that for each $i$ and each data category $x_t^i \neq$ <MASK>,

$$q(x_t^i | \boldsymbol{x}_{t+1}) = \sum_{\tilde{\boldsymbol{x}}_t} q(\tilde{\boldsymbol{x}}_t | \boldsymbol{x}_{t+1}) \cdot q(x_t^i | \tilde{\boldsymbol{x}}_t, \boldsymbol{x}_{t+1}). \tag{22}$$

If $i \in J$, then both the LHS of the above equation and $q(x_t^i | \tilde{\boldsymbol{x}}_t, \boldsymbol{x}_{t+1})$ equals one if and only if $x_t^i = x_{t+1}^i$. Therefore, the result holds trivially.

Next, if $i \in I$, denote $I_{\setminus i} := I \setminus \{i\}$, Equation (22) is simplified to

$$q(x_t^i | \boldsymbol{x}_{t+1}) = \sum_{\tilde{\boldsymbol{x}}_t} q(\tilde{\boldsymbol{x}}_t | \boldsymbol{x}_{t+1}) \cdot q(x_t^i | \tilde{\boldsymbol{x}}_t, \boldsymbol{x}_{t+1}),$$

$$= \sum_{\tilde{x}_t^i} \sum_{\tilde{\boldsymbol{x}}_t^{I_{\setminus i}}} q(\tilde{x}_t^i, \tilde{\boldsymbol{x}}_t^{I_{\setminus i}} | \boldsymbol{x}_{t+1}) \cdot q(x_t^i | \tilde{x}_t^i, x_{t+1}^i),$$

$$= q(\tilde{X}_t^i = x_t^i | \boldsymbol{x}_{t+1}) \cdot q(x_t^i | \tilde{X}_t^i = x_t^i, x_{t+1}^i),$$

$$= q(\tilde{X}_t^i = x_t^i | \boldsymbol{x}_{t+1}) \cdot \frac{\alpha_{t+1} - \alpha_t}{\alpha_{t+1}}.$$

Therefore, we have

$$q(\tilde{X}_t^i = x_t^i | \boldsymbol{x}_{t+1}) = \frac{1}{Z} \cdot q(X_t^i = x_t^i | \boldsymbol{x}_{t+1}), \text{ where } Z = \sum_{x_t^i \neq \texttt{<MASK>}} q(X_t^i = x_t^i | \boldsymbol{x}_{t+1}).$$

$\square$

## B  Relation Between $\mathcal{L}(\mathbf{V}; p_{\text{tar}}, p_{\text{est}})$ and Matrix Scaling

The matrix scaling problem gives a matrix $A$ as input and asks for diagonal 'scaling' matrices $X$ and $Y$ such that $XAY$ is doubly stochastic (its row and column sums are all one). More generally, target row and column sum vectors $r$ and $c$ are provided and need not contain only ones. The solvability of this problem for positive matrices was established by Sinkhorn (1964), and its algorithms (sometimes called iterative proportional fitting), generalizations, and numerous applications have been studied thoroughly (Kalantari & Khachiyan, 1993; Ruschendorf, 1995; Allen-Zhu et al., 2017); see (Idel, 2016) for a review. Taking the multidimensional generalization of the problem and interpreting the tensor as a (unnormalized) probability distribution yields the connection to our problem, with the target sums being the univariate marginal distributions.

## C  Parameterizing Discrete Copulas by Odds Ratios

We start by formally defining odds ratios.

**Definition 2** (Rudas (2018)). Let $p$ be a distribution over variables $\boldsymbol{X}$ each taking values in $\{0, 1\}$. For a partition of $\boldsymbol{X}$ into sets $\boldsymbol{A}$ and $\boldsymbol{B}$, the *conditional odds ratio* of variables $\boldsymbol{A}$ conditioned on the assignment $\boldsymbol{B} = \boldsymbol{b}$ is

$$\text{COR}_p(\boldsymbol{A} | \boldsymbol{B} = \boldsymbol{b}) = \frac{\prod_{\boldsymbol{a} \in s} p(\boldsymbol{a}, \boldsymbol{b})}{\prod_{\boldsymbol{a} \in d} p(\boldsymbol{a}, \boldsymbol{b})}$$

where $s$ is the set of assignments to $\boldsymbol{A}$ whose parity is the same as the number of variables in $\boldsymbol{A}$, and $d$ is the set of assignments whose parity is different.

In the case of more than two categories per variable, $\text{COR}_p(\boldsymbol{A} | \boldsymbol{B} = \boldsymbol{b})$ can generalized further to be a set of similarly defined ratios (see, e.g., Rudas (2018)). Together the set of all conditional odds ratios $\text{COR}_p(\boldsymbol{A} | \boldsymbol{B} = \boldsymbol{b})$ for partitions of $\boldsymbol{X}$ into sets $\boldsymbol{A}$ and $\boldsymbol{B}$ with $|\boldsymbol{A}| \geq 2$, completely specifies the association among the variables in the joint distribution $p$, as established by the following theorem.

**Theorem 2** (Rudas (2018)). *Let $q$ and $r$ be positive probability distributions on a the set of variables $\boldsymbol{X}$ each taking values in $\{0, 1, \ldots, k\}$. Then there exists a unique probability distribution $p$ such that $p$ has the same univariate marginal distributions as $q$, that is, for all $i$*

$$p(x_i) = q(x_i),$$

*and $p$ has the same copula as $q$, that is for all partitions of $\boldsymbol{X}$ into sets $\boldsymbol{A}$ and $\boldsymbol{B}$ with $|\boldsymbol{A}| \geq 2$,*

$$COR_p(\boldsymbol{A} | \boldsymbol{B} = \boldsymbol{b}) = COR_r(\boldsymbol{A} | \boldsymbol{B} = \boldsymbol{b}).$$

*Proof.* This follows from (Rudas, 2018, Theorem 10.2) by taking the descending set to contain the empty set and all singletons (and the ascending set, its complement). $\square$

Theorem 2 shows how any distribution $p$ can be viewed as combining independent marginal distributions (i.e., from $r$) and odds ratios (i.e., from $q$). Such a combination has desirable properties. For example, in the case of two variables with possibly many categories, it has been shown that among all distributions with the same margins as $r$, the distribution $p$ minimizes the KL-divergence to $q$ (Geenens, 2020, Theorem 6.2), i.e. that $p$ is the information projection of $q$ onto the set of distributions with the margins of $r$.

# D  UNBIASED UNIVARIATE MARGINALS FROM DISCRETE DIFFUSION MODELS

In this section, we show that when their respective training losses are minimized, discrete-time and continuous-time discrete diffusion models recover the true univariate marginals.

**Discrete-Time Diffusion Models.**  Discrete-time diffusion models (Austin et al., 2021) are trained to maximize the ELBO between the forward joint distribution $p(\boldsymbol{x}_0)q(\boldsymbol{x}_{1:T}|\boldsymbol{x}_0)$, where $p(\boldsymbol{x}_0)$ is the data distribution, and the reverse joint distribution $p_\theta(\boldsymbol{x}_{0:T})$. The ELBO can be simplified to

$$\mathbb{E}_q\left[\log\frac{p(\boldsymbol{x}_T)}{q(\boldsymbol{x}_T)} + \sum_{t=1}^T \log\frac{p_\theta(\boldsymbol{x}_{t-1}|\boldsymbol{x}_t)}{q(\boldsymbol{x}_{t-1}|\boldsymbol{x}_t)} + \log p(\boldsymbol{x}_0)\right].$$

Assume that $p_\theta(\boldsymbol{x}_{t-1}|\boldsymbol{x}_t)$ encodes fully-factorized distribution, the above objective can be simplified as

$$\sum_{t=1}^T \sum_i q(x_{t-1}^i|\boldsymbol{x}_t)\log\frac{p_\theta(x_{t-1}^i|\boldsymbol{x}_t)}{q(x_{t-1}^i|\boldsymbol{x}_t)} + \mathbb{E}_q\left[\log\frac{p(\boldsymbol{x}_T)}{q(\boldsymbol{x}_T)} + \log p(\boldsymbol{x}_0)\right],$$

where the second term is independent to $p_\theta$. From the first term of the above formula, we can conclude that the ELBO objective is maximized when $p_\theta(x_{t-1}^i|\boldsymbol{x}_t) = q(x_{t-1}^i|\boldsymbol{x}_t)$ for every $t$ and every $i$.

**Continuous-Time Diffusion Models.**  As described in Section 2, many continuous-time diffusion models learn to approximate the likelihood ratio (defined as $s_\theta(\boldsymbol{x}_t, \boldsymbol{x}_t'; t)$) at all noise levels $t \in [0, T]$:

$$s_\theta(\boldsymbol{x}_t, \boldsymbol{x}_t'; t) := \frac{q(\mathbf{X}_t = \boldsymbol{x}_t')}{q(\mathbf{X}_t = \boldsymbol{x}_t)}.$$

Specifically, Lou et al. (2024); Meng et al. (2022) directly parameterize a neural network to approximate the likelihood ratios, and Sun et al. (2022) approximates the likelihood ratios with the conditional distributions $p_\theta(X_t^i|\boldsymbol{x}_t^{\backslash i})$ ($\forall i, t$).

For each $\boldsymbol{x}_t$, since there are exponentially many possible $\boldsymbol{x}_t'$, it is infeasible to have a neural network to directly model the likelihood ratio for all pairs of $(\boldsymbol{x}_t, \boldsymbol{x}_t')$. Instead, they focus on $(\boldsymbol{x}_t, \boldsymbol{x}_t')$ pairs where $\boldsymbol{x}_t$ and $\boldsymbol{x}_t'$ are only different in one single variable, i.e., their Hamming distance is one. For example, in Lou et al. (2024), they represent $s_\theta$ as $s_\theta(\boldsymbol{x}_t, y_t^i; t, i)$, which computes the likelihood ratio between $\boldsymbol{x}_t$ and $\boldsymbol{x}_t' = \{\boldsymbol{x}_t^{\backslash i}, y_t^i\}$. $s_\theta$ is trained by minimizing the following objective:

$$\mathbb{E}_{t, \boldsymbol{x}_t \sim q(\mathbf{X}_t)}\left[\sum_i \sum_{y_t^i \neq x_t^i} w_t\left(s_\theta(\boldsymbol{x}_t, y_t^i; t, i) - \frac{q(\mathbf{X}_t = \{\boldsymbol{x}_t^{\backslash i}, y_t^i\})}{q(\mathbf{X}_t = \boldsymbol{x}_t)}\log s_\theta(\boldsymbol{x}_t, y_t^i; t, i)\right)\right],$$

where $\{w_t\}_t$ are positive weights. When the above objective is minimized, $s_\theta$ recovers the correct likelihood ratios:

$$\forall i, t, \ s_\theta(\boldsymbol{x}_t, y_t^i; t, i) = \frac{q(\mathbf{X}_t = \{\boldsymbol{x}_t^{\backslash i}, y_t^i\})}{q(\mathbf{X}_t = \boldsymbol{x}_t)}. \tag{23}$$

At inference time, continuous-time discrete diffusion models select a list of time steps $0 < t_0 < \cdots < t_k = T$ to sample from: first sample from the prior $p(\mathbf{X}_{t_k})$ and then sample recursively from $\{p_\theta(\boldsymbol{x}_{t_{i-1}}|\boldsymbol{x}_{t_i})\}_{i=1}^k$, where $p_\theta(\boldsymbol{x}_{t_{i-1}}|\boldsymbol{x}_{t_i})$ is obtained from $s_\theta(\boldsymbol{x}_t, y_t^i; t, i)$ in an indirect manner. Specifically, assume $\frac{dp(\boldsymbol{x}_t)}{dt} = Q \cdot p(\boldsymbol{x}_t)$, we have[3]

$$q(\boldsymbol{x}_{t_{i-1}}|\boldsymbol{x}_{t_i}) = q(\boldsymbol{x}_{t_i}|\boldsymbol{x}_{t_{i-1}}) \cdot \frac{q(\boldsymbol{x}_{t_{i-1}})}{q(\boldsymbol{x}_{t_i})},$$

$$= q(\boldsymbol{x}_{t_i}|\boldsymbol{x}_{t_{i-1}}) \cdot \left(\sum_{\boldsymbol{x}}\exp(-\Delta t \cdot Q)(\boldsymbol{x}_{t_{i-1}}, \boldsymbol{x}) \cdot \frac{q(\mathbf{X}_{t_i} = \boldsymbol{x})}{q(\mathbf{X}_{t_i} = \boldsymbol{x}_{t_i})}\right),$$

---

[3]This argument largely follows Theorem 4.1 in Lou et al. (2024). We include it for the sake of completeness.

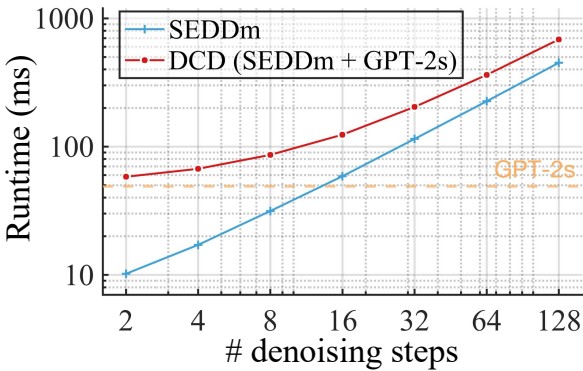

Figure 7: Sampling time of DCD and its two base models with 2 to 128 denoising steps.

where $\Delta t := t_i - t_{i-1}$ and $\exp(-\Delta t \cdot Q)(\boldsymbol{x}_{t_{i-1}}, \boldsymbol{x})$ denotes the product of $\exp(-\Delta t \cdot Q)(x_{t_{i-1}}^j, x^j)$, the $x_{t_{i-1}}^j$-th row and $x^j$-th column of $\exp(-\Delta t \cdot Q)$.

Plug in Equation (23), we can compute the marginal of $x_{t_{i-1}}^j$ (i.e., $p_\theta(x_{t_{i-1}}^j | \boldsymbol{x}_{t_i})$) following

$$q(X_{t_{i-1}}^j = y | \boldsymbol{x}_{t_i}) \propto q(\boldsymbol{x}_{t_i} | \boldsymbol{x}_{t_{i-1}}) \cdot \left( \sum_{y'} \exp(-\Delta t \cdot Q)(y, y') \cdot s_\theta(\boldsymbol{x}_{t_i}, y'; t_i, j) \right),$$

$$= \exp(\Delta t \cdot Q)(y, x_{t_i}^j) \cdot \left( \sum_{y'} \exp(-\Delta t \cdot Q)(y, y') \cdot s_\theta(\boldsymbol{x}_{t_i}, y'; t_i, j) \right).$$

Therefore, if $s_\theta$ perfectly learns the likelihood ratios between inputs with Hamming distance at most one, then the correct marginals $q(x_{t_{i-1}}^j | \boldsymbol{x}_{t_i})$ can be computed using $s_\theta$.

# E  IMPLEMENTATION DETAILS OF DCD

We describe details about the "autoregressive" version of DCD introduced in Section 5.3. According to Section 5.3, the first $(T-t-1)/T$ portion of the tokens in $\boldsymbol{x}_{t+1}$ are unmasked. At step $t$, we only need to additionally unmask the tokens spanning the $(T-t-1)/T$ to $(T-t)/T$ fraction of the sequence $\boldsymbol{x}_t$. We do this by caching the keys and values generated by the attention layers of tokens generated in previous denoising steps. So at step $t$, we will have the KV-caches of the first $(T-t-1)/T$ fraction of tokens. As a result, the computational cost for running the autoregressive Transformer is independent of the number of denoising steps.

---

**Algorithm 2** DCD with Autoregressive Copula Models and Using Autoregressive Sampling

---

1: **Inputs:** a diffusion model $p_{\mathrm{dm}}$, an autoregressive model $p_{\mathrm{ar}}$, number of time steps $T$, sequence length $L$
2: **Outputs:** a sample $\boldsymbol{x}_0$ from the discrete diffusion model augmented by the autoregressive model
3: **Initialize:** Sample $\boldsymbol{x}_T$ from the prior noise distribution $p(\mathbf{X}_T)$
4: **for** $t = T-1$ **to** 0
5:    $i_{\min}, i_{\max} = \frac{L}{T} \cdot (T-t-1), \frac{L}{T} \cdot (T-t)$ (w.l.o.g. assume $L$ is divisible by $T$)
6:    Compute $\{p_{\mathrm{dm}}(\tilde{X}_t^i | \boldsymbol{x}_{t+1})\}_i$ and $\{p_{\mathrm{dm}}(\tilde{X}_t^i | \boldsymbol{x}_{t+1}^{<i})\}_i$ for each $i \in [i_{\min}, i_{\max})$ using the diffusion model
7:    Compute $\mathbf{V}[i, \tilde{x}_t^i] = \log p_{\mathrm{dm}}(\tilde{x}_t^i | \boldsymbol{x}_{t+1}) - \log p_{\mathrm{dm}}(\tilde{x}_t^i | \boldsymbol{x}_{t+1}^{<i})$ ($\forall i \in [i_{\min}, i_{\max}), \tilde{x}_t^i$)
8:    $\boldsymbol{x}_t \leftarrow \boldsymbol{x}_{t+1}$
9:    **for** $i = i_{\min}$ **to** $i_{\max} - 1$
10:    └ Sample $x_t^i$ from $\hat{p}(x_t^i) \propto p_{\mathrm{ar}}(x_t^i | \boldsymbol{x}_t^{<i}) \cdot \prod_i \exp(\mathbf{V}[i, x_t^i])$ and store it to $\boldsymbol{x}_t$

---

**Additional Runtime Analysis.**  Figure 7 displays the generation time per sample for $\mathrm{SEDD_M}$, $\mathrm{GPT\text{-}2_S}$, and DCD. When the number of denoising steps is small, the computation cost of running $\mathrm{GPT\text{-}2_S}$ dominates the total runtime of DCD. However, as the number of denoising steps increases, this cost is amortized because, with KV-caching, the total computation cost for running $\mathrm{GPT\text{-}2_S}$ stays constant.

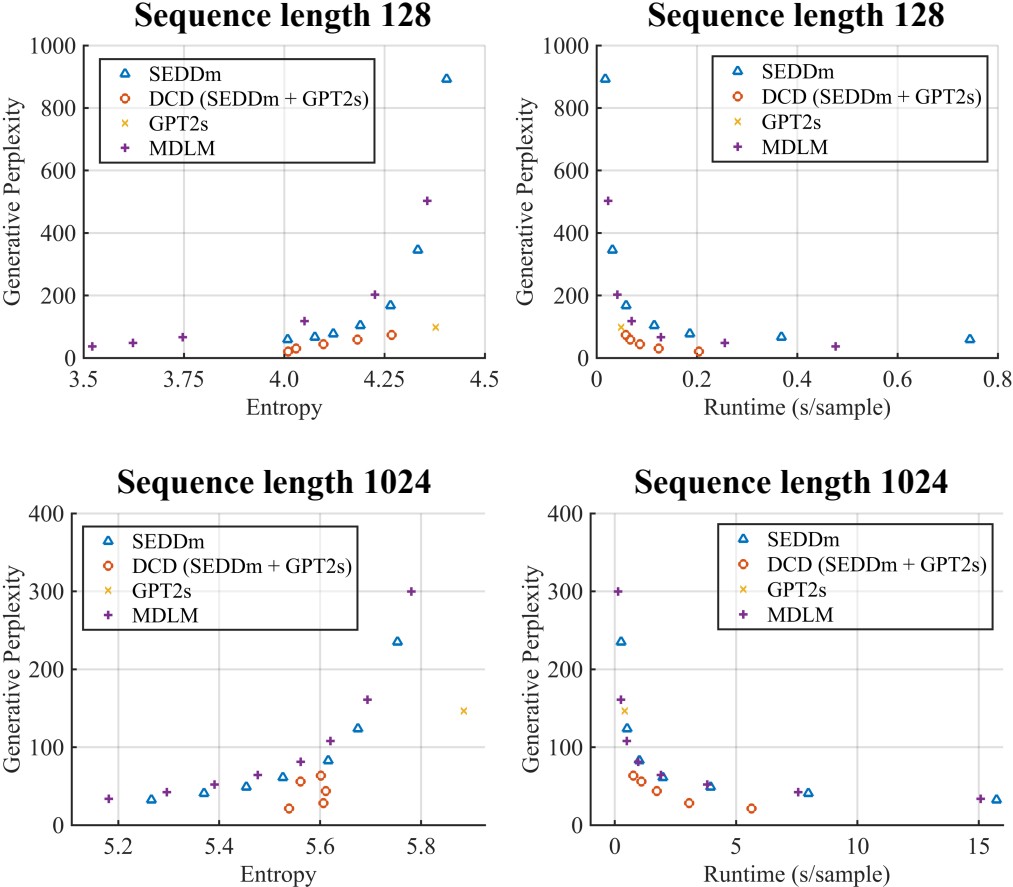

Figure 8: Comparison between generative perplexity (↓), diversity (measured by sentence entropy; ↑), and runtime (↓) of DCD with baselines.

## F ADDITIONAL UNCONDITIONAL GENERATION EXPERIMENTS

To better understand the relation between quality (measured by generative perplexity), diversity (measured by sentence entropy[4]), and speed for DCD and its baselines. Specifically, we run the more efficient version of DCD described in the last paragraph of Section 5.3 and Appendix E to generate text sequences of lengths 128 and 1024. In addition to SEDD and GPT2, the two base models used by DCD, we compare them with MDLM (Sahoo et al., 2024), a more recent discrete diffusion model that is more efficient than SEDD. Note that DCD can use any discrete diffusion model as its base model.

First, we compare the sample time and the generative perplexity (the second and the fourth sub-plot in Figure 8). Compared to SEDD, GPT, and MDLM, DCD consistently achieves better generative perplexity given a fixed runtime constraint. It also requires less time to achieve a desired perplexity value.

Additionally, we compare the perplexity and diversity of the generated text sequences. Following community standards, we adopt the sentence entropy to measure the diversity of generated text. Specifically, the entropy of each text sequence is the entropy of its token frequency distribution, and the final sentence entropy is the average entropy over all generated sequences. The desired behavior is to have low generative perplexity and high sentence entropy (which means high diversity). Results are shown in the table below and Figure 8's first and third sub-plot. Compared to the two discrete diffusion models (SEDD and MDLM), DCD achieves better generative perplexity under the same

---

[4]The sentence entropy of a sequence is the entropy of its token frequency distribution. The reported number is averaged across all samples.

entropy, which offers a better perplexity-diversity tradeoff. Compared to the autoregressive GPT model, although the entropy of DCD is lower, it achieves better generative perplexity with slightly worse entropy.

# G    ADDITIONAL EXPERIMENTAL DETAILS

This section provides additional details of the experiments.

## G.1    UNCONDITIONAL TEXT GENERATION

**SEDD.**    We adopt the SEDD-medium model with 320M non-embedding parameters trained on OpenWebText. The model is accessed through HuggingFace: `https://huggingface.co/louaaron/sedd-medium`. We follow the original paper (Lou et al., 2024) and use the log-linear noise schedule $\sigma(t) = -\log(1 - (1 - \epsilon t))$, which leads to the forward transition probabilities ($0 \leq s < t \leq T$):

$$q(\boldsymbol{x}_t | \boldsymbol{x}_s) := \mathrm{Cat}(\boldsymbol{x}_t; \exp(\sigma(t - s) \cdot Q) \cdot \boldsymbol{x}_s).$$

The absorbing mask forward noising process is used. The corresponding transition rate matrix is

$$Q := \begin{bmatrix} -1 & 0 & \cdots & 0 & 0 \\ 0 & -1 & \cdots & 0 & 0 \\ \vdots & \vdots & \ddots & \vdots & \vdots \\ 0 & 0 & \cdots & -1 & 0 \\ 1 & 1 & \cdots & 1 & 0 \end{bmatrix},$$

where the last category is `<MASK>`.

**GPT.**    The GPT-2-small model is obtained from HuggingFace: `https://huggingface.co/openai-community/gpt2`.

**DCD.**    We implement DCD by combining $\mathrm{SEDD_M}$ and $\mathrm{GPT\text{-}2_S}$ following the steps in Algorithm 1. In line 8, instead of masking tokens independently, we group chunks of 8 tokens together and mask/unmask them with the same probability given the noise schedule (i.e., $\alpha_t/\alpha_{t+1}$ as shown in Prop. 5).

## G.2    CONDITIONAL TEXT GENERATION

**MAUVE Score.**    We adopt the MAUVE implementation available in the Python package `evaluate`. We use the default hyperparameters established by the original paper (Pillutla et al., 2021), which is also the default used by the package. We found that the number of samples and the number of samples given a fixed prompt influenced the score. Therefore, we randomly selected the 2,000 prompts and generated 5 samples for each prompt for all methods.

**Detailed Runtime Analysis.**    As shown in Algorithm 1, in each denoising step of DCD, we need to run the discrete diffusion model twice: first to compute $\{p(\tilde{X}_t^i | \boldsymbol{x}_{t+1})\}_i$ and next to compute $\{p(\tilde{X}_t^i | \boldsymbol{x}_{t+1}^{<i})\}_i$ by applying causal attention masks to the same denoising neural network given that it is based on the Transformer architecture. Next, as discussed in Appendix E, the total runtime consumed by the autoregressive model remains constant across different numbers of denoising steps thanks to the KV-caching mechanism. Therefore, the runtime of DCD will be dominated by the computation cost of the autoregressive model with only a few denoising steps. As the number of denoising steps increases, the runtime of the autoregressive model will be amortized and the total computation cost will be dominated by the cost to evaluate the diffusion model.

**SSD-LM.**    SSD-LM (Han et al., 2023) is a semi-autoregressive model that uses techniques from discrete diffusion models to predict/denoise chunks of sequences in an autoregressive manner. Specifically, given a predefined chunk size, SSD-LM diffuses tokens in each chunk one by one conditioned on all previous chunks. As a result, the model is semi-autoregressive and cannot see suffix prompts.

While the official implementation on GitHub (`https://github.com/xhan77/ssd-lm`) only allows conditioning on tokens in previous prompts, we improved their code to also allow conditioning on tokens in the current chunk that is being diffused. Specifically, we replace the diffusion model's input corresponding to the prompt tokens with the ground truth token embeddings.

We followed the original paper to choose a chunk size of 32 and use top-p sampling with $p = 0.95$. The remaining hyperparameters are kept as default.

### G.3 ANTIBODY SEQUENCE INFILLING

**Detailed Task Description.** The adopted antibodies with an immunoglobulin G (IgG) format, which comprises a heavy (H) chain and a light (L) chain. Each chain has three complementarity determining regions (CDRs) that are crucial toward the binding affinity to the target antigen.

**Training NOS-D.** We use the training script as well as the dataset provided in the official GitHub repo of NOS-D (`https://github.com/ngruver/NOS`). The model is trained with 50 epochs using the default settings (e.g., learning rate and its schedule).

**Training GPT.** We use the same dataset provided in the repository of NOS-D and use the GPT implementation from `https://github.com/karpathy/nanoGPT/tree/master`. The GPT model has 6 layers, an embedding size of 512, and 16 attention heads. The model is trained for 10 epochs with the default settings in the nanoGPT repository.

**DCD.** When implementing DCD for the antibody sequence infilling task, we add an additional scaling factor to the coefficients in $\mathbf{V}$. That is, $\mathbf{V}$ is updated in line 6 of Algorithm 1 following

$$\forall i, \tilde{x}_t^i, \ \mathbf{V}[i, \tilde{x}_t^i] = \beta \cdot \left( \log p_{\mathrm{dm}}(\tilde{x}_t^i | \boldsymbol{x}_{t+1}) - \log p_{\mathrm{dm}}(\tilde{x}_t^i | \boldsymbol{x}_{t+1}^{<i}) \right),$$

where we set $\beta = 0.1$ for this task. We note that $\beta = 1$ works well for the language modeling tasks. The need to choose a smaller $\beta$ in this task may be caused by the fact that the dataset and the models are much smaller and are more prone to overfitting.

## H ADDITIONAL RELATED WORK

We briefly review a class of related works that perform (semi-)autoregressive diffusion, which is weakly related to our work since we also "combine" discrete diffusion models with autoregressive models. Specifically, Wu et al. (2023); Chen et al. (2024) perform diffusion in an autoregressive manner by allowing the noising schedule to be variable-dependent. Variables at the beginning are kept unchanged at small $t$s and are corrupted only when $t$ is close to $T$; variables at the end are corrupted in the first few time steps. During sampling where $t$ moves from $T$ to 0, initial variables/tokens are first denoised, followed by later tokens. This allows the diffusion model to perform "autoregressive denoising".

## I ADDITIONAL TEXT SAMPLES

We provide randomly selected unconditional samples in Figure 9 and 10 and conditional samples in Figure 11 and 12.

…, DHHS, Dion Todd of Detroit, Detroit Tigers team players Marcus Johnson of Detroit Lions, Minnesota Vikings team players Troy Polamalu of Detroit Tigers, mellow lines for opposing teams, but they do not share a common mentality.

"At the end of the day, if you're just grouping me but you also come for the tram stop, everybody plays by the rules," Kalim said of Johnson. "If you start to play offense, you're going to make mistakes. We don't just give them time because we don't think they're going to win. We don't just teach them how to take care …

Both 1-2 range panels are available for 3 hours of ongoing use (25 burning power consumption plus periodic gas combustion cycle). Also included moisture protection from agricultural insects and biological spills of greenhouse gases due to different costs of livestock sit idle in the energy cycle of so many nations.

The high yield yields from large, agricultural biofuels rely entirely on larger projections placed together by country by government into 2020. Such predictions assume an 1 billion tonne increase in capacity to 5 billion tonnes per year six billion years from now then consumption for nearly all full-time, undertaking- Stage 3 + 2.0 exponential growth.

… acquisition jack. Add this to the negotiations, and the place starts going down. Veteran grocers like State Farm have halted stocking their own toll booths and warnings because they fear getting squeezed out by major retailers. Meanwhile, stores like Wal-Mart have reduced shelf space by as much as 8.6 percent. Wal-Mart Stores Canada has been the most profitable Wal-Mart store chain in Canada (green grocer WalMart Stores Canadian now accounts for nearly a third of sales, up from just 1 percent in 1996). Meanwhile Wal-Mart continues to aggressively sell Canadian goods. Since 1995, Wal-Mart Stores Canada has more than tripled the volume and …

I find myself divided. Like I was growing up in this world. I saw my dad constantly being obsessed about faith, constantly being remembered in my mind his name. My dad thought that I wasn't going to grow up to say myself, if I tried to make my family happy they wouldn't believe me anymore. Every time I looked in my eyes I thought I was crazy, but I thought I was my brother. I was worried I would always feel jealous of my father and slowly, I started to think about family. I found a family where everyone ruled me alone towards the end, in hell. Family was always there, it was

flap: Now you can lay your hands on a wavy pattern without touching the nerves, or if you fancy you can lay your arms on an occluded specialty? Any hypothesis relied from those vainly generalized action.[303] Reconciling the patient's single hairs with multiple llings stipulated that a single story could be shortened in half, but in fact lengthened in less flexible forms. If, however, little succeeded at the scholarly step, that compromise was subtlety in his notes, provided he lacked enough hairs to lie straight down. He could even weave rings for his harp—especially wheat—but there was …

Figure 9: Randomly selected unconditional samples from DCD ($\text{SEDD}_\text{M}+\text{GPT-2}_\text{S}$) with 4 denoising steps.

March if they can overcome federal complaints.

Farmers say they have acquiesced to pressure from U.S. agribusiness giants to cut back on pesticides, while environmentalists say regulators allege a lack of oversight by officials.

"We won't tolerate anything bad," said Rutko Guerra, spokesman for Cornell University Extension. "It's a clear conflict of interest."

Cornell University Extension estimates about 400 pesticides are sold through the city each year in violation of the Public Health Act and other state laws, prompting over $49 million in fines — the largest ever levied by a public university.

Each time components are created a buffer is created. This buffer holds the components and should be updated whenever they touch on the screen. The buffer foundation passes every buffer value from the component to the component's buffer.

After the component has been created a markers is placed inside the marker stating which colors are used in the new paint direction. During the paint direction there are three modes: launch to draw pixels, around draw so baccarat will appear misaligned, and lowdraw so baccarat will appear perfectly aligned.

… in on their autobiographical stories of realizing happiness.The Stimulant Prophecy was an important spiritual awakening that occurred during the 1950s, 1960s and 1970s. It awakened believers to cultivate spiritual fortitude as well as resolve conflicts and lead to more successful relationships.According to Tages Jephzei (The Stimulant Prophecy), this event ultimately caused Muslims to develop compassion for one another by their communal experiences.It also fueled support for incongruity in mainline Muslim societies and gave hope for physical cleansings.In 1976, Tages Jephzei published his book The Stimulant Prophecy: Understanding Muslims …

…would proper address regulations affecting this country." He recalled how he supported Deputy Minority Leader Nancy Pelosi's (D-Calif.) efforts to explore Russian meddling in the 2016 election: "When Nancy Pelosi [D-Calif.] spoke to me, he wanted to consider Russian interference in our election." Kennedy also added that he believes a new law requiring commercial polluters to disclose their emissions during emissions tests will help clean up the air: "[I] hope that the EPA will follow through with two years of programs that are going to reduce [during gas] emissions in some form or another," Kennedy told reporters.

Figure 10: Randomly selected unconditional samples from DCD ($SEDD_M + GPT-2_S$) with 32 denoising steps.

… are hear the exhortations "We ask of everyone to speak the voice of God" and "we ask to be loved **for the game and what they wanted to do next for the series.**" In fact, PS3 announcement executives confirmed this month that Sony Pictures Entertainment plans to release PS Vita versions of Sony PlayStation Classics PlayStationGS, The Last of Us, Ratchet, Square **did not come up with the " revolutionary " idea that would warrant a new entry for the PlayStation 3. Speaking in an interview**, it was revealed that Square Enix felt Square Enix could not offer a few new plugins without seeing Square Enix make Square Enix's " visionary Battle …

… in the American version, and Warner Bros. added Nobuo Ukiura of Miyazaki Animation to directing on character design. **A large team of writers handled the script. The game's** story was developed by Kouki Watanabe in Chouki no Namco Europe, and Fujikyo Pictures Entertainment released it theatrically on May 25, 1999 in North America, followed **with an expanded edition in November of that year. It was also adapted into manga and an original video animation series. Due to low sales**, Warner Bros. suffered widespread cancellation due to lack of revenue. Warner Bros. also shut it down due to its failure to …

They save as many enemies as you can through Chrono Trigger Online missions, unlock quests in missions missions, unlock special quests, **having a higher difficulty than those found in the rest of the game.** These include boss & combat objectives. Chrono Trigger Online contains one Chrono Trigger EP with unlockable Chrono Trigger ARC girls.

This is the first patch which implements the PTZ **system, is carried over directly from Valkyira Chronicles. During missions, players select each unit using a top @**-left position. They determine their unit type, which determines their ability and the size of their field of vision. They can only activate …

The Final Fantasia in Japanese / **Media.Vision for the PlayStation Portable. Released in January 2011 in Japan, it** debuted as a novel while on hold in the North East Stand. It garnered reviews for its breathtaking narrative, disappointing plot, and creepy characters. A fourth graphic novel is in the first game **and follows the " Nameless ", a penal military unit serving the nation of Gallia during the Second Europan War.**

… scenery was was composed in short animation. When the game ended early on Mikami Sakura was drawn by Makoto Masui.. A large team of writers handled the script. The game's soundtrack lasted around 12 hours and Sugiyama Shogarashi, Makoto Masuyama and Kyoko Takamura included Takme Ibara. The music theme was originally released in 2009, **with an expanded edition in November of that year. It was also adapted into manga and an original video animation series. Due to low sales, the** game release was delayed to three weeks in spring 2011. Following its final release on May 23rd 2013. …

Figure 11: Randomly selected conditional samples from DCD (SEDD$_M$ + GPT-2$_S$) with 4 denoising steps. Prompt texts are bolded and in blue.

… every character has. You can choose combat situation best suit the **character. To learn Battle Potentials, each character has a unique** skills makes them invaluable. One of Potentials best suit the character is Point squirrel on the map, the character Leda can use skills like " Star Wars Matchmaker", the character Jaden **can activate " Direct Command " and move around the battlefield without depleting his Action Point gauge, the character Reila can shift** to melee objects to send morgues (so more reliable), Mira can change her story situation to battle nweire battle to ward strategise, a " Command Pointcher " is …

… this State building institution. We could be build a defensive system **to the United States Arsenal in this city ( Little Rock ).** This system seem really feasible and good. The name of the City that would to this scenario go by the Rocky Mountain Sound as the Academy as well is MADISON FIREWRIGHT NASHA ROCK. **-John M Harrel Telegram, January 31, 1861**
**The item was intended simply as a piece of news, but** it also served as an "opportunity" for the U.S. Fortresses on Little Rock. Setting aside the exception of the basisicks' menansi slogan, it was all …

… on the air at the end of those years run in early **1923. An original design for the society called The Darling of** the American for Being Unnamed was put forth on the air at the end of those years held early 1925. The unnamed pursuits of the American were previously documented by History Magazine called St. Luke's **Society for the Propagation of the Gospel in its November 1919 exhibition.**
**Religious @-@ themed books include** The Red Book, The Hidden Voyage, an opera which was written on behalf of John Ford and produced under the contract of the Protestant revival organization, The Evangelical Fund (without Contemplation) …

letter to city Evans noted in more details reminded Christine Barker to **supervise the household, and to give both her mother and sister** complete authority to their development. (See Evansdone & Sullivan) 79. Christine Barker continued her adult life but when she reached ends of age, during which time her big sister Ruth had died unexpectedly **of a heart attack. Barker was unable to pursue her art to any significant extent following her sister's death, as all** of her parents perished and she lacked the discipline, learning needed to be as a professional age. (See Evanssic) 80. Although moving art was a lifetime profession for Christine Barker, bear …

Pool : At Mumbai airport Shivaji Park. Women Technical girls **under @-@ 17 women's team competed in Confederation of** Asian Football's premier youth competition. 2011 year-13 results : :U 13 medal : NW15 qualification : A pool order : Of the 155 young women, five girls had to be narrowed **from an initial pool of 49 young women. Two girls from the SOS Children ' s Village Bakoteh** were chosen for the USC and two girls from the Meijer 's Village Bakoteh. The remaining Meijer girl was selected for the opening ceremony. After the AU's teams

Figure 12: Randomly selected conditional samples from DCD (SEDD$_M$+GPT-2$_S$) with 32 denoising steps. Prompt texts are bolded and in blue.

