# OpenReview forum: "Discrete Copula Diffusion"
_ICLR.cc/2025/Conference — ICLR 2025 Poster_

### Official Review · Reviewer_5yGc · 2024-11-01

**Soundness:** 3
**Presentation:** 3
**Contribution:** 3
**Rating:** 6
**Confidence:** 4

**Summary:**

The paper proposes using copulas to enhance the approximation accuracy of the backward process in masked diffusion models. Specifically, it shows that the true posterior $q(x_t|x_{t+1})$ can be decomposed into a latent auxiliary variable model $q(x_t|x_{t+1}) = \sum_{\tilde{x}_t} q(\tilde{x}_t | x_{t+1}) q(x_t|\tilde{x}_t, x_{t+1})$, in which $q(x_t|\tilde{x}_t, x_{t+1})$ has a closed form and $q(\tilde{x}_t | x_{t+1})$ can be approximated using a pre-trained auto-regressive model. To further enhance modelling capabilities, the paper suggests learning an I-projection of $q(\tilde{x}_t | x_{t+1})$, so that the univariate marginals align with the true data marginals. Experimental results demonstrate that this approach reduces the number of sampling steps.

**Strengths:**

The paper is well-motivated and  proposes a theorectically sounded framework to improve masked discrete diffusion models. Decomposing the true posterior into a latent auxiliary variable model is a very interesting observation.

**Weaknesses:**

I have two main concerns:

1. Although the proposed method reduces the number of sampling steps, the number of function evaluations (NFE) appears to be quite large. Unlike discrete diffusion, which requires only one NFE per step, each step in the proposed method involves evaluating the autoregressive model, the diffusion model, and solving a convex optimization problem. This suggests that the computational cost is significantly higher than that of discrete diffusion and autoregressive models.
2.  As shown in Eq.6, the paper uses a pre-trained auto-regressive model to approximate $q(\tilde{x}_t | x_{t+1})$. However this is a biased approximation. This bias can impact the overall fidelity of the backward process, potentially leading to discrepancies in the modeled distribution.

**Questions:**

- In line with weakness 1, could you add a plot in Figure 3 showing the relationship between running time and perplexity, and include the running time for GPT-2 models as well?
- In addressing weakness 2, could you consider a variant model that solely uses Equation 5 for denoising, without applying the copula? This would provide insight into how well the autoregressive model approximates the conditional distribution of the data.
- Moreover, one interesting idea to address weakness 2, could be to use generative marginal models [1] to learn p(x_0), and the conditional in eq.5 has close form. Specifically,  p(x^I | x^J) could be computed using Bayes’ rule: p(x^I , x^J) / p(x^J). The trade-off here is that while enerative marginal models may not be as expressive as autoregressive models, they provide exact computations, offering a different balance between model power and precision.
- Proposition 1 is confusing. There is no model distribution in Equation 2, so it's not obvious how relaxing the independent denoising assumption would reduce the negative ELBO. Additionally, in line 727 of the appendix, could you explain in more detail how the inequality holds?
- Could you also apply the proposed discrete copula diffusion to MDLM [2,3] and present results for running time versus perplexity? This would help address concerns about the method’s broader applicability across all masked diffusion models. Additionally, it would be valuable to explore whether improvements in the autoregressive model lead to better performance in the proposed method, which raises an interesting question worth investigating.
- Some suggestion in tmers of writing
    - On line 293, instead of stating "it can be decomposed...," it would be clearer to explicitly write out: $q(x_t|x_{t+1}) = \sum_{\tilde{x_t}} q(\tilde{x_t} | x_{t+1})q(x_t|\tilde{x_t}, x_{t+1})$
    - Could you add a paragraph in the main text discussing the limitations of the proposed method and potential directions for future work? Including this would help motivate readers to explore further improvements and applications.

[1] Liu, Sulin, Peter J. Ramadge, and Ryan P. Adams. "Generative Marginalization Models." *arXiv preprint arXiv:2310.12920* (2023).

[2] Shi, Jiaxin, et al. "Simplified and Generalized Masked Diffusion for Discrete Data." *arXiv preprint arXiv:2406.04329* (2024).

[3] Sahoo, Subham Sekhar, et al. "Simple and Effective Masked Diffusion Language Models." *arXiv preprint arXiv:2406.07524* (2024).

---

> ### Author Response · Authors · 2024-11-22
>
> We thank the reviewer for their constructive feedback.
>
> > Weakness 1: computation cost of DCD
>
> > Could you also apply the proposed discrete copula diffusion to MDLM [2,3] and present results for running time versus perplexity?
>
> We conducted additional unconditional text generation experiments to facilitate a fair comparison of the quality (measured by generative perplexity), diversity (measured by sentence entropy), and speed for DCD and its baselines. Specifically, we run the more efficient version of DCD described in the last paragraph of Section 5.3 and Appendix E to generate text sequences of lengths 128 and 1024. In addition to SEDD and GPT2, the two base models used by DCD, we compare them with MDLM, a more recent discrete diffusion model that is more efficient than SEDD. Note that DCD can use any discrete diffusion model as its base model.
>
> Below are tables containing results for sequence length 128. For better visualization, we have plotted the results for both length 128 and length 1024 in Appendix F of the revised paper. First, we compare the sample time and the generative perplexity (the second and the fourth sub-plot in Figure 7). Compared to SEDD, GPT, and MDLM, DCD consistently achieves better generative perplexity given a fixed runtime constraint. It also requires less time to achieve a desired perplexity value.
>
> | # steps         | 2       | 4       | 8       | 16       | 32       | 64       | 128       | 256       |
> |----------------|---------|---------|---------|---------|---------|---------|---------|---------|
> | SEDD Entropy   | 4.4874  | 4.4046  | 4.3330  | 4.2647  | 4.1895  | 4.1223  | 4.0761  | 4.0084  |
> | SEDD Perplexity| 2958.24 | 892.49  | 345.83  | 167.98  | 104.09  | 77.79   | 67.05   | 59.13   |
> | SEDD Runtime| 0.0102 | 0.0171 | 0.0315 | 0.0587 | 0.1148 | 0.1858 | 0.3684 | 0.7445 |
> |MDLM Entropy| 4.5002 | 4.4399 | 4.3562 | 4.2262 | 4.0507 | 3.7471 | 3.6229 | 3.5217 |
> |MDLM Perplexity| 3531.30 | 1373.38 | 503.03 | 202.65 | 117.93 | 66.36 | 48.13 | 36.95 |
> |MDLM Runtime| 0.0116 | 0.0164 | 0.0229 | 0.0413 | 0.0698 | 0.1280 | 0.2557 | 0.4766 |
> |DCD Entropy | 4.2677 | 4.1824 | 4.0976 | 4.0297 | 4.0099 | - | - | - |
> |DCD Perplexity| 73.48 | 58.73 | 43.95 | 30.15 | 20.87 | - | - | - |
> |DCD Runtime| 0.0582 | 0.0671 | 0.0862 | 0.1238 | 0.2038 | - | - | - |
>
> Additionally, we compare the perplexity and diversity of the generated text sequences. Following community standards, we adopt the sentence entropy to measure the diversity of generated text. Specifically, the entropy of each text sequence is the entropy of its token frequency distribution, and the final sentence entropy is the average entropy over all generated sequences. The desired behavior is to have low generative perplexity and high sentence entropy (which means high diversity). Results are shown in the table below and Figure 7's first and third sub-plot. Compared to the two discrete diffusion models (SEDD and MDLM), DCD achieves better generative perplexity under the same entropy, which offers a better perplexity-diversity tradeoff. Compared to the autoregressive GPT model, although the entropy of DCD is lower, it achieves better generative perplexity with slightly worse entropy.
>
> We note that this version of DCD changes the ordering for unmasking tokens, which can be a limitation for certain tasks (e.g., when we desire random unmasking). As described in the paper, this variant is also used in the text infilling and antibody infilling tasks, so this does not limit the model’s ability to do infilling/imputation, but it could affect certain applications such as controlling the generation process with techniques similar to classifier-based/-free guidance. However, the additional results demonstrate that DCD can efficiently generate high-quality samples. They could also motivate future work to further improve DCD or to solve the inter-variable dependency problem of discrete diffusion models in a more principled way.

---

> > ### Author Response · Authors · 2024-11-22
> >
> > > In addressing weakness 2, could you consider a variant model that solely uses Equation 5 for denoising, without applying the copula?
> >
> > We thank the reviewer for the suggestion. However, we think there might be a misunderstanding about Equation 5. Solely using Equation 5 for denoising refers to only using the (autoregressive) copula model, as Section 5.1 discusses how to use autoregressive models as copula models for discrete diffusion with the absorbing mask noising strategy. As discussed in Section 3, the challenge of modeling variable dependencies is more severe in the few-step diffusion case, where the model has the “unmask” at least two tokens in each denoising step. As shown in Section 6.1 and the additional results in the response to the previous question, when given sufficient denoising steps, discrete diffusion models can achieve better generative perplexity despite being much slower and also having worse diversity. The goal of DCD is to highlight this variable dependency problem and to provide a preliminary solution to achieve a better tradeoff between speed and sample quality.
> >
> > Regarding the second weakness, we agree with the reviewer that autoregressive copula models are biased as they cannot observe future tokens, which is discussed in Section 5.1 (the paragraph starting at line 319). We show in Section 4 (e.g., Proposition 2) that the I-projected distribution is always a better approximation of the target distribution regardless of the biases in the copula model. Although for autoregressive copula models, we can only approximate the I-projected distribution, the hope is that by combining generative models with different biases (i.e., diffusion models lack inter-variable dependencies while autoregressive models capture such dependencies but cannot condition on future tokens), we can get the best from both worlds to have a model that can (i) generate higher-quality samples and (ii) perform tasks beyond next-token prediction. Our original experiments as well as the new experiments performed in the rebuttal period demonstrate that DCD can generate better (lower generative perplexity and higher diversity) samples in less time compared to strong diffusion model baselines. It is also able to perform tasks such as sequence infilling, which is infeasible for autoregressive models.
> >
> > > Moreover, one interesting idea to address weakness 2, could be to use generative marginal models [1] to learn p(x_0), and the conditional in eq.5 has close form.
> >
> > We thank the reviewer for their insightful suggestion. Generative marginal models can be a very nice fit as a copula model, and they can be directly applied in Algorithm 1 to replace autoregressive copula models. DCD is a general algorithm framework that allows the application of different types of copula models to improve the few-step generation performance of discrete diffusion models. We believe combining generative marginal models is a very promising future direction.
> >
> > > Proposition 1 is confusing. There is no model distribution in Equation 2, so it's not obvious how relaxing the independent denoising assumption would reduce the negative ELBO.
> >
> > Equation 2 is the lower bound of the ELBO that can be achieved by *any* diffusion model (i.e., any parameterization of the denoising neural network) with the independent denoising assumption described in the first line of Proposition 1. In addition to the independent denoising assumption, this lower bound depends on the data distribution and the number of denoising steps. Therefore, under the independent denoising assumption, the only way to reduce the lower bound is by adding more denoising steps.
> >
> > If we can relax the independent denoising assumption, then the lower bound of the negative ELBO is the data entropy $H(p(\mathbf{X}_{0}))$.

---

> > > ### Author Response · Authors · 2024-11-22
> > >
> > > > Additionally, in line 727 of the appendix, could you explain in more detail how the inequality holds?
> > >
> > > The third term of Equation 11 corresponds to the data entropy term in the lower bound, and the second term of Equation 11 can be simplified into the sum of total correlations assuming that the marginals of the denoising distributions are perfectly learned (the paragraph of line 718). Finally, the first term of Equation 11 is greater than or equal to zero and can be absorbed by the $\geq$ symbol in line 726. We have updated the proof in the revised paper to include more intermediate steps.
> > >
> > > > On line 293, instead of stating "it can be decomposed...," it would be clearer to explicitly write out: ….
> > >
> > > We thank the reviewer for their suggestion. We have modified the sentence in the revised paper.
> > >
> > > > Could you add a paragraph in the main text discussing the limitations of the proposed method and potential directions for future work?
> > >
> > > There are several limitations to the DCD framework. First, in addition to a discrete diffusion model, it requires another copula model, which may require additional training for certain applications. Second, although the I-projected distribution is guaranteed as a better estimate of the target distribution, the I-projection step often needs to be approximated in practice. While DCD achieves better performance in language generation and antibody infilling, DCD may not always achieve better performance than its base models. Finally, although DCD requires fewer denoising steps, the computation cost of each step is higher than in discrete diffusion models. Therefore, DCD may not always provide a speedup compared to the base diffusion model. However, DCD points out the inter-dependency modeling problem in existing discrete diffusion models and opens up the possibility of combining different types of generative models for better overall performance. Another potential future direction is to directly train (variants of) diffusion models that capture inter-variable dependencies.
> > >
> > > We have included the discussion in Section 7 of the revised paper.

---

> > > ### Comment · Reviewer_5yGc · 2024-11-25
> > >
> > > Thanks for your detailed response. Most of my concerns have been addressed. I have one question regarding my previous concern: "could you consider a variant model that solely uses Equation 5 for denoising, without applying the copula".
> > >
> > > I agree that the main contribution of the paper lies in highlighting the importance of capturing variable dependency in discrete diffusion models. However, for me, the most interesting and insightful aspect is Proposition 5. Without it, developing a copula model on top of pre-trained AR models would not be possible. Therefore, I still believe that including an experiment demonstrating the transformation of an AR model into a diffusion model using Proposition 5, without additional training, would be highly promising and provide valuable insights.
> > >
> > > I am genuinely curious whether this approach would work or not. Please don’t worry about the results—they won’t affect my evaluation of your contribution. I already think the paper is solid. If you have conducted related experiments, it would be great to hear the outcome. Otherwise, sharing your thoughts on its feasibility would also be appreciated.
> > >
> > > Additionally, I believe there are alternative ways to implicitly introduce variable dependency into discrete diffusion models. For instance, you could parameterize the model as an EBM: $p(x_0 | x_t) \propto p_{diff}(x_0 | x_t) p_{AR}(x_0 )$ where $p_{diff}(x_0 | x_t)$ denotes the pretraned diffusion models and $p_{AR}(x_0)$ is the pretrained AR model. To sample from $p(x_0 | x_t)$, you can use importance sampling as in [1]. I think this gives you an alternative way to introduce variable dependency. It would be great if you could include it in your discussion. It is okay that you don't have experimental comparisons because [1] is a concurrent work. But it would be great to have some discussion in the related work or future direction.
> > >
> > > [1] Guo, Wei, et al. "Plug-and-Play Controllable Generation for Discrete Masked Models." arXiv preprint arXiv:2410.02143 (2024).

---

> > > > ### Author Response · Authors · 2024-11-25
> > > >
> > > > We thank the reviewer for acknowledging the improvements and quality of the paper. In the following, we discuss the reviewer's additional comments.
> > > >
> > > > > Therefore, I still believe that including an experiment demonstrating the transformation of an AR model into a diffusion model using Proposition 5, without additional training, would be highly promising and provide valuable insights.
> > > >
> > > > We thank the reviewer for their insightful comment. Following the suggestions, we evaluate GPT-2 (the adopted autoregressive model) as if it were a diffusion model following Proposition 5. Specifically, starting from the sequence with all masked tokens, at each denoising step, we use the autoregressive model to unmask tokens following $q ( x_{t} \vert x_{t+1} )$.
> > > > According to Proposition 5, this can be decomposed into first sampling all tokens $x̃_{t}$ following the autoregressive distribution (Eq. (5)), and then masking each token independently with a probability specified by the noising strategy. Note that when sampling $x̃_{t}$, for those already unmasked tokens (i.e., tokens not masked in $x_{t+1}$), we keep the original token according to Equation (5). In the following, we report unconditional text generation performance on sequences of length 128. We use the same setting in Section 6.1 and evaluate both the generative perplexity and the sentence entropy (a measure of diversity). Results are shown in the following.
> > > >
> > > > | # Denoising Steps    | 16       | 32       | 64       | 128      |
> > > > |-----------------------|----------|----------|----------|----------|
> > > > | Generative Perplexity | 361.01   | 780.07   | 1962.85  | 4490.40  |
> > > > | Sentence Entropy      | 4.4876   | 4.5259   | 4.5713   | 4.6022   |
> > > >
> > > > As shown in the above table, with just the autoregressive copula model, the performance is much worse than GPT-2 with autoregressive sampling and DCD. This suggests that while the autoregressive model serves as a good copula model in DCD, using only this (autoregressive) copula model will not lead to good generation performance as it is *fundamentally biased* — it cannot condition on future context. Specifically, when adopting the absorbing mask noising process, the input $\mathbf{x}_{t}$ to each denoising step is a sequence where tokens are randomly masked. While discrete diffusion models are trained to sample masked tokens based on *all currently unmasked tokens*, autoregressive models can only condition on previous unmasked tokens when generating any token. This bias causes the bad generation performance of autoregressive models in this “diffusion model generation setting”, and the performance becomes worse when more denoising steps are used since there will on average be more unmasked tokens in future tokens.
> > > >
> > > > Although autoregressive models are not good distributions for the “diffusion model setting”, they are still very good copula models since they accurately capture the dependencies between different masked tokens in any denoising step. Through the I-projection procedure, DCD uses the accurate univariate marginals from discrete diffusion models, while only using the copula captured by autoregressive copula models. Therefore, the combined model (DCD) achieves better performance than both of its base models.
> > > >
> > > > We will run additional experiments for sequences of length 1024 and incorporate the discussion into the paper once they are completed.
> > > >
> > > > > To sample from p(x0|xt), you can use importance sampling as in [1]. I think this gives you an alternative way to introduce variable dependency.
> > > >
> > > > We thank the reviewer for mentioning the relevant concurrent work [1]. We agree that the DCD framework can be adapted to using other types of generative models as copula models. [1] provides one nice example by showing that energy-based models can be used to model inter-variable dependencies for discrete diffusion models. We have added the discussion into Section 7 of the revised paper.

---

> > > > > ### Author Response · Authors · 2024-11-27
> > > > >
> > > > > Dear reviewer 5yGc,
> > > > >
> > > > > Thank you again for the insightful comments. Since the discussion period is about to end, could you kindly check if our response addressed your concerns? We are happy to answer any follow-up questions and concerns.
> > > > >
> > > > > Best,
> > > > >
> > > > > Authors

---

### Official Review · Reviewer_YU9j · 2024-11-03

**Soundness:** 2
**Presentation:** 3
**Contribution:** 3
**Rating:** 6
**Confidence:** 3

**Summary:**

Traditional Discrete Diffusion models suffer from severe degradation in sampling quality when the number of sampling steps is reduced. In contrast, continuous Diffusion Models don't have this issue. This paper first points out that this is because discrete diffusion models cannot capture the dependencies between output variables at each denoising step. To address this problem, the paper proposes incorporating another deep generative model, named the copula model. The copula model can be trained independently and only needs to be incorporated during the sampling process of the Discrete diffusion model. Experiments demonstrate that the proposed method can significantly improve the sampling speed of discrete diffusion models.

**Strengths:**

1. This paper addresses a very important issue: how to improve the sampling efficiency of diffusion models (reducing sampling steps while maintaining sampling quality) is a crucial topic. While this problem has been well solved in continuous diffusion models, it remains unresolved in discrete diffusion models.

2. This paper is well-written, and I particularly appreciate its insightful illustrations. For example, Figure 1 provides an excellent clarification of the existing problems in traditional Discrete Diffusion models.

3. The paper provides solid theoretical analysis for the proposed method.

4. The proposed method doesn't require modifications to the diffusion model during training, thus offering excellent scalability.

**Weaknesses:**

1. Since this paper requires training an additional (new) generative model, the training requirements of the proposed method might be higher. Considering that Autoregressive methods are already very well-suited for generative modeling of discrete data, I am uncertain if this paper feels like using a complex model to improve a relatively simple model.

2. Using perplexity metrics alone is not sufficient. For example, we expect the generated text to be sufficiently diverse, but perplexity metrics struggle to reflect the diversity of generated samples.

3. Most experiments were conducted with a generation length of 128, which does not reflect real-world scenarios, e.g., with a length of 1024.

**Questions:**

1. I have questions about the results in Figure 4. The figure gives me the impression that if we further increase the number of denoising steps, SEDD's performance could still improve, while DCD's results can't improve anymore. I would like to know how the performance comparison between the two would look if the number of denoising steps were further increased.

2. Since I'm not very familiar with synthetic text data, I'm unsure whether perplexity is an appropriate metric for measuring the quality of generated data (especially the ability to restore the ground-truth data distribution). Are there any other metrics to measure if the dependencies between different variables are correctly captured? For other types of discrete data, such as tabular data, there are more metrics available to measure the quality of generated data (such as metrics for correlation between different dimensions). I would like to know how the method proposed in this paper performs on these types of data.

3. Can the authors provide a more detailed comparison of the computational costs and training requirements between their proposed method and existing approaches?  As demonstrated in Figure 5, your methods look much slower.

---

> ### Author Response · Authors · 2024-11-22
>
> We thank the reviewer for their constructive feedback.
>
> > Since this paper requires training an additional (new) generative model, the training requirements of the proposed method might be higher. Considering that Autoregressive methods are already very well-suited for generative modeling of discrete data, I am uncertain if this paper feels like using a complex model to improve a relatively simple model.
>
> For tasks such as language modeling, DCD can directly take a pretrained diffusion model and a pretrained autoregressive model and combine them at inference time, and hence we do not need to train an additional generative model.
>
> One main benefit of DCD is to improve the performance of discrete diffusion models on tasks that can be done by diffusion models but not by the copula model (e.g., autoregressive models). For example, while autoregressive models like GPTs have better unconditioned generation performance than most discrete diffusion models, they cannot perform tasks such as text infilling as they cannot condition on suffix prompts. With DCD, we can benefit from both the expressiveness of autoregressive models and the built-in capability of diffusion models to perform text infilling. This strength is elaborated in the experiments shown in Sections 6.2 and 6.3.
>
> For downstream tasks that are feasible for both diffusion models and autoregressive models (e.g., unconditional generation), the goal of this paper is to demonstrate a simple yet effective way to combine both pretrained models to achieve even better generative performance. As shown in Figure 3, by combining GPT and SEDD, DCD can achieve better overall generative perplexity.
>
> > Using perplexity metrics alone is not sufficient. For example, we expect the generated text to be sufficiently diverse, but perplexity metrics struggle to reflect the diversity of generated samples.
>
> > Most experiments were conducted with a generation length of 128, which does not reflect real-world scenarios, e.g., with a length of 1024.
>
> > Can the authors provide a more detailed comparison of the computational costs and training requirements between their proposed method and existing approaches?
>
> We have conducted additional experiments in unconditional text generation to better understand the relation between quality (measured by generative perplexity), diversity (measured by sentence entropy), and speed for DCD and its baselines. Specifically, we run the more efficient version of DCD described in the last paragraph of Section 5.3 and Appendix E to generate text sequences of lengths 128 and 1024. In addition to SEDD and GPT2, the two base models used by DCD, we compare them with MDLM, a more recent discrete diffusion model that is more efficient than SEDD. Note that DCD can use any discrete diffusion model as its base model.
>
> Below are tables containing results for sequence length 128. For better visualization, we have plotted the results for both length 128 and length 1024 in Appendix G of the revised paper. First, we compare the sample time and the generative perplexity (the second and the fourth sub-plot in Figure 7). Compared to SEDD, GPT, and MDLM, DCD consistently achieves better generative perplexity given a fixed runtime constraint. It also requires less time to achieve a desired perplexity value.
>
> | # steps         | 2       | 4       | 8       | 16       | 32       | 64       | 128       | 256       |
> |----------------|---------|---------|---------|---------|---------|---------|---------|---------|
> | SEDD Entropy   | 4.4874  | 4.4046  | 4.3330  | 4.2647  | 4.1895  | 4.1223  | 4.0761  | 4.0084  |
> | SEDD Perplexity| 2958.24 | 892.49  | 345.83  | 167.98  | 104.09  | 77.79   | 67.05   | 59.13   |
> | SEDD Runtime| 0.0102 | 0.0171 | 0.0315 | 0.0587 | 0.1148 | 0.1858 | 0.3684 | 0.7445 |
> |MDLM Entropy| 4.5002 | 4.4399 | 4.3562 | 4.2262 | 4.0507 | 3.7471 | 3.6229 | 3.5217 |
> |MDLM Perplexity| 3531.30 | 1373.38 | 503.03 | 202.65 | 117.93 | 66.36 | 48.13 | 36.95 |
> |MDLM Runtime| 0.0116 | 0.0164 | 0.0229 | 0.0413 | 0.0698 | 0.1280 | 0.2557 | 0.4766 |
> |DCD Entropy | 4.2677 | 4.1824 | 4.0976 | 4.0297 | 4.0099 | - | - | - |
> |DCD Perplexity| 73.48 | 58.73 | 43.95 | 30.15 | 20.87 | - | - | - |
> |DCD Runtime| 0.0582 | 0.0671 | 0.0862 | 0.1238 | 0.2038 | - | - | - |
>
> (To be continued in the next comment.)

---

> > ### Author Response · Authors · 2024-11-22
> >
> > Additionally, we compare the perplexity and diversity of the generated text sequences. Following community standards, we adopt the sentence entropy to measure the diversity of generated text. Specifically, the entropy of each text sequence is the entropy of its token frequency distribution, and the final sentence entropy is the average entropy over all generated sequences. The desired behavior is to have low generative perplexity and high sentence entropy (which means high diversity). Results are shown in the table below and Figure 7's first and third sub-plot. Compared to the two discrete diffusion models (SEDD and MDLM), DCD achieves better generative perplexity under the same entropy, which offers a better perplexity-diversity tradeoff. Compared to the autoregressive GPT model, although the entropy of DCD is lower, it achieves better generative perplexity with slightly worse entropy.
> >
> > For text infilling tasks, we follow the community standard and test sequences of length 128. We also used more challenging masks while prior work focused mostly on filling the 50% in the middle.
> >
> > We note that this version of DCD changes the ordering for unmasking tokens, which can be a limitation for certain tasks (e.g., when we desire random unmasking). As described in the paper, this variant is also used in the text infilling and antibody infilling tasks, so this does not limit the model’s ability to do infilling/imputation, but it could affect certain applications such as controlling the generation process with techniques similar to classifier-based/-free guidance. However, the additional results demonstrate that DCD can efficiently generate high-quality samples. They could also motivate future work to further improve DCD or to solve the inter-variable dependency problem of discrete diffusion models in a more principled way.
> >
> > > I have questions about the results in Figure 4. The figure gives me the impression that if we further increase the number of denoising steps, SEDD's performance could still improve, while DCD's results can't improve anymore.
> >
> > Results in Figure 4 contain sample text from SEDD and DCD to show intuitively that SEDD cannot produce high-quality samples with few denoising steps while DCD can achieve good sample quality in as few as 4 steps. Further increasing the number of denoising steps of SEDD will not help much since with 256 steps to generate sequences of length 128, only one token will be unmasked in each denoising step. Additionally, as shown in Figure 3, the performance of DCD improves consistently as we increase the number of denoising steps.
> >
> > >  Are there any other metrics to measure if the dependencies between different variables are correctly captured? For other types of discrete data, such as tabular data, there are more metrics available to measure the quality of generated data (such as metrics for correlation between different dimensions).
> >
> > The antibody sequence infilling experiment in Section 6.3 provides a more direct justification that DCD captures variable dependency better. The evaluation metric here is the sequence recovery rate, the accuracy of the imputed amino acids. Since there is a unique answer to each infilling sequence, the sequence recovery rate has a strong positive correlation with the correctness of the captured variable dependency.

---

> > > ### Author Response · Authors · 2024-11-27
> > >
> > > Dear reviewer YU9j,
> > >
> > > Thank you again for the insightful comments. Since the discussion period is about to end, could you kindly check if our response addressed your concerns? We are happy to answer any follow-up questions and concerns.
> > >
> > > Best,
> > >
> > > Authors

---

### Official Review · Reviewer_bAgJ · 2024-11-05

**Soundness:** 3
**Presentation:** 2
**Contribution:** 2
**Rating:** 5
**Confidence:** 5

**Summary:**

This paper proposes the Discrete Copula Diffusion (DCD) method. It addresses the inability of discrete diffusion models to capture inter-variable dependencies during denoising, which hampers their few-step generation performance. DCD combines a discrete diffusion model with an autoregressive copula model at inference. By integrating the univariate marginals from the diffusion model and the dependencies captured by the copula model, it approximates the true denoising distribution. Experiments in unconditional and conditional text generation and antibody sequence infilling demonstrate that DCD outperforms individual models, achieving better results with significantly fewer denoising steps, thus emphasizing the importance of modeling inter-variable dependencies in discrete diffusion.

**Strengths:**

1. Originality: the paper proposes an interesting approach of combining a discrete diffusion model with a copula model to address the issue of capturing inter-variable dependencies in discrete diffusion, which is a relatively new direction.

2. Quality: the paper provides a formal analysis of the problem in discrete diffusion models, showing the limitations of the independent denoising assumption and how to improve it with the copula model.

3. Clarity: The paper is well-structured, and explanations of concepts like copula models, I-projection, and the noising process in discrete diffusion are detailed and understandable, making it easy to follow the flow of ideas.

**Weaknesses:**

My main concerns are about the experiment setup and results.

1. Reasonability of assumption: from Sec 4, the motivation is that the diffusion model learns better the marginal distributions, while copula (which is AR model) captures correlation but may be more biased in marginals. I think it's better to show empirical evidence for this assumption.

2. Computational cost: this is my main concern. Could the author provide a simple complexity analysis for sampling time? According to Alg 1, I think line 5 requires twice forward computation with bidirectional and causal attention, and line 7 (eq 6) even needs another computation. As shown in Fig, this process is extremely slow.

3. I suggest having another plot like Fig 3 but making the x-axis the wall clock time, so it shows efficiency directly. As indicated by Fig 5, I guess the practical acceleration is limited.

4. An important baseline is missing. I suggest also comparing DCD against MDLM [1] where the cache sampler can also significantly reduce timestep and accelerate the sampling speed.

[1] Sahoo, Subham Sekhar, et al. "Simple and Effective Masked Diffusion Language Models."

5. Evaluation protocols are too synthetic. The paper just studies 128-length text generation and shows the significance, but typical studies are all done on longer sequences such as 1024 length. Besides, it's critical to also report diversity scores for the generations. Generative PPL is known to suffer from mode collapse generation without consideration for diversity.

**Questions:**

1. Is the combined distribution $\hat{p}$ (Eq 3) a normalized distribution? Otherwise, the notation should be proportionate.

2. Could we compute the likelihood (or bound) for DCD model? Is so, it will be good to have a perplexity metric which is also a common benchmark; if not, what's the challenge for it?

---

> ### Author Response · Authors · 2024-11-22
>
> We thank the reviewer for their constructive feedback.
>
> > Reasonability of assumption: from Sec 4, the motivation is that the diffusion model learns better the marginal distributions, while copula (which is AR model) captures correlation but may be more biased in marginals. I think it's better to show empirical evidence for this assumption.
>
> Combining discrete diffusion models with autoregressive copula models is beneficial even when the (implicit) marginal distributions learned by the autoregressive are better. Consider the case of text infilling, where we are given both prefix prompts and suffix prompts. Even if the autoregressive model learns the perfect distribution, due to its inability to condition on suffix prompts, the conditioned marginal distributions will be much less accurate than that of the diffusion model, which is able to condition on all given tokens.
>
> Therefore, combining discrete diffusion models and autoregressive models in a principled way can produce hybrid models that are better than each individual model: from the diffusion model’s perspective, the autoregressive model adds inter-variable dependencies; from the autoregressive model’s perspective, the diffusion model allows it to condition on suffix prompts, extending beyond just autoregressive generation.
>
> >  I suggest also comparing DCD against MDLM [1] where the cache sampler can also significantly reduce timestep and accelerate the sampling speed
> > The paper just studies 128-length text generation and shows the significance, but typical studies are all done on longer sequences such as 1024 length.
> > Besides, it's critical to also report diversity scores for the generations.
>
> We have conducted additional experiments in unconditional text generation to better understand the relation between quality (measured by generative perplexity), diversity (measured by sentence entropy), and speed for DCD and its baselines. Specifically, we run the more efficient version of DCD described in the last paragraph of Section 5.3 and Appendix E to generate text sequences of lengths 128 and 1024. In addition to SEDD and GPT2, the two base models used by DCD, we compare them with MDLM, a more recent discrete diffusion model that is more efficient than SEDD. Note that DCD can use any discrete diffusion model as its base model.
>
> Below are tables containing results for sequence length 128. For better visualization, we have plotted the results for both length 128 and length 1024 in Appendix F of the revised paper. First, we compare the sample time and the generative perplexity (the second and the fourth sub-plot in Figure 7). Compared to SEDD, GPT, and MDLM, DCD consistently achieves better generative perplexity given a fixed runtime constraint. It also requires less time to achieve a desired perplexity value.
>
> | # steps         | 2       | 4       | 8       | 16       | 32       | 64       | 128       | 256       |
> |----------------|---------|---------|---------|---------|---------|---------|---------|---------|
> | SEDD Entropy   | 4.4874  | 4.4046  | 4.3330  | 4.2647  | 4.1895  | 4.1223  | 4.0761  | 4.0084  |
> | SEDD Perplexity| 2958.24 | 892.49  | 345.83  | 167.98  | 104.09  | 77.79   | 67.05   | 59.13   |
> | SEDD Runtime| 0.0102 | 0.0171 | 0.0315 | 0.0587 | 0.1148 | 0.1858 | 0.3684 | 0.7445 |
> |MDLM Entropy| 4.5002 | 4.4399 | 4.3562 | 4.2262 | 4.0507 | 3.7471 | 3.6229 | 3.5217 |
> |MDLM Perplexity| 3531.30 | 1373.38 | 503.03 | 202.65 | 117.93 | 66.36 | 48.13 | 36.95 |
> |MDLM Runtime| 0.0116 | 0.0164 | 0.0229 | 0.0413 | 0.0698 | 0.1280 | 0.2557 | 0.4766 |
> |DCD Entropy | 4.2677 | 4.1824 | 4.0976 | 4.0297 | 4.0099 | - | - | - |
> |DCD Perplexity| 73.48 | 58.73 | 43.95 | 30.15 | 20.87 | - | - | - |
> |DCD Runtime| 0.0582 | 0.0671 | 0.0862 | 0.1238 | 0.2038 | - | - | - |
>
> Additionally, we compare the perplexity and diversity of the generated text sequences. Following community standards, we adopt the sentence entropy to measure the diversity of generated text. Specifically, the entropy of each text sequence is the entropy of its token frequency distribution, and the final sentence entropy is the average entropy over all generated sequences. The desired behavior is to have low generative perplexity and high sentence entropy (which means high diversity). Results are shown in the table below and Figure 7's first and third sub-plot. Compared to the two discrete diffusion models (SEDD and MDLM), DCD achieves better generative perplexity under the same entropy, which offers a better perplexity-diversity tradeoff. Compared to the autoregressive GPT model, although the entropy of DCD is lower, it achieves better generative perplexity with slightly worse entropy.
>
> (To be continued in the next comment.)

---

> > ### Author Response · Authors · 2024-11-22
> >
> > We note that this version of DCD changes the ordering for unmasking tokens, which can be a limitation for certain tasks (e.g., when we desire random unmasking). As described in the paper, this variant is also used in the text infilling and antibody infilling tasks, so this does not limit the model’s ability to do infilling/imputation, but it could affect certain applications such as controlling the generation process with techniques similar to classifier-based/-free guidance. However, the additional results demonstrate that DCD can efficiently generate high-quality samples. They could also motivate future work to further improve DCD or to solve the inter-variable dependency problem of discrete diffusion models in a more principled way.
> >
> > > According to Alg 1, I think line 5 requires twice forward computation with bidirectional and causal attention, and line 7 (eq 6) even needs another computation.
> >
> > In addition to the above empirical results, we conceptually discuss the efficiency of DCD. For DCD with N diffusion steps, 2*N calls of the diffusion model are needed (line 6 in Algorithm 1). Additionally, there is a constant cost of running the autoregressive model to sample all tokens once (since we can use KV-caching; described in line 396). Therefore, as shown in Figure 5, the runtime of DCD with N diffusion steps is 2 times the base diffusion model plus a constant overhead from the autoregressive model, which is amortized as the number of diffusion steps goes up. In comparison with the 8 to 32 times saving in number of denoising steps, the computation cost is only 2 to 4 times larger per denoising step.
> >
> > > Is the combined distribution $\hat{p}$ (Eq 3) a normalized distribution?
> >
> > It is a normalized distribution by definition. For any set of coefficients V, we can normalize them such that $\hat{p}$ is a normalized distribution. As an example, the solution to Eq. (4) offers a normalized distribution $\hat{p}$.
> >
> > > Could we compute the likelihood (or bound) for DCD model? Is so, it will be good to have a perplexity metric which is also a common benchmark; if not, what's the challenge for it?
> >
> > It is challenging to provide a tight bound on the likelihood of DCD due to the autoregressive nature of the adopted GPT model. According to the MDLM paper, approximating the ELBO of discrete diffusion models requires computing the conditional probability of masked tokens given *any* partially-mask sequence. However, autoregressive models can only compute conditional likelihoods when all previous tokens are provided.

---

> > > ### Author Response · Authors · 2024-11-27
> > >
> > > Dear reviewer bAgJ,
> > >
> > > Thank you again for the insightful comments. Since the discussion period is about to end, could you kindly check if our response addressed your concerns? We are happy to answer any follow-up questions and concerns.
> > >
> > > Best,
> > >
> > > Authors

---

### Official Review · Reviewer_nyBv · 2024-11-12

**Soundness:** 3
**Presentation:** 2
**Contribution:** 3
**Rating:** 6
**Confidence:** 4

**Summary:**

This paper studies the dependency issue of multivariables in each step of discrete diffusion model.  The paper claims that discrete diffusion model requires many steps because that each step of the diffusion model cannot fully capture the required inter-variable dependency. To model the dependency among these variables, the author proposes an interesting setting, that given both accurate estimation of marginal distribution and approximated estimation of full distribution, how to obtain a much accurate estimation of the full distribution. An optimization algorithm,  I-projection, is proposed to solve the problem.
To inject inter-variable dependency for each diffusion step, the author proposes to estimate the full distribution with an fixed-ordering autoregressive model, and then approximate each step's target distribution of certain marginalization of the autoregressive model. Many approximations are proposed to achieve the final "correction" step.

**Strengths:**

1. The proposed issue of inter-variable dependency of generated elements within diffusion model is an interesting observation and a valid issue.

2. The proposed mathematical problem that finding a better estimation of full distribution given both marginal distributions and inaccurately estimated full distribution is a very interesting problem that can combines two type of methods together. The author shows that diffusion model and autoregressive model can be combined under the proposed setting.

3. The author proposed a good realization of the proposed combination: how to estimate an approximated target distribution with inter-variable dependency using autoregressive model, and how to find the I-project's solution to derive the much accurate estimation of the target distribution. (Under absorbing

4. In experiment, the author shows that the method can achieve better performance than all baselines even given a few diffusion steps.

**Weaknesses:**

1. Lack an detailed explanation of why the inter-dependency issue is only for discrete diffusion model but not for continuous diffusion model, which can affect the soundness.
2. Section 2 line 121 mentioned that continuous-time case the objective is optimizing likelihood ratio. This is not fully correct, given [1] shows that the SEDD's objective can be exactly derived from ELBO setting. The author also lacks reference for its continuous extension.
3. Lack a clear definition of "copula", while it appears many times in many different ways.
4. Section 5.1 needs a major revision: it is hard to read while the described equations are not that hard. Some suggestions: define the each variable's meaning before putting them into equations. like X-tilde, which I don't understand before read equation (5) in detail. Also, give an illustration figure to describe the setting of each variable shown in equation (5) under absorbing state case. Most importantly, describing your easy-to-understand intuition and main thought before throwing all equations that hide your thoughts deeply.
5. Need certain consistency, for example, equation (5) and (6) 's last part: one use element case while the other use combined case. Notation/equation should be consistent. I also don't like the notation of I and J which does not have the t+1 subscript but only represent the case for t+1 time. These notation/equation issues make the paper hard to follow.
6. The proposed analysis has many approximations that can introduce huge approximation error, in both copula distribution of I-projection. I personally think the error within the copula distribution can be huge, given that the autoregressive method is based on the inherent ordering of the sequence (like text), while the conditioned part of equation (6) clearly needs more unmasked tokens. Ideally, one would like to train an "autoregressive diffusion" model proposed by Emiel Hoogeboom [2], which can make the ordering issue less problematic. The ideal copula model should work for any combination of J and I ordering for any time step t, which is a huge requirement. I suggest the author think about how to quantify the influence of these approximations in copula model and the I-projection stage.
7. Another limitation is that the current copula model only works for absorbing state diffusion model. Which is not ideal and not widely used for many other domains such as graphs given it does not address many symmetry problems.







[1] Zhao, L., Ding, X., Yu, L., & Akoglu, L. (2024). Improving and Unifying Discrete&Continuous-time Discrete Denoising Diffusion. arXiv preprint arXiv:2402.03701.
[2] Hoogeboom, E., Gritsenko, A. A., Bastings, J., Poole, B., Berg, R. V. D., & Salimans, T. (2021). Autoregressive diffusion models. arXiv preprint arXiv:2110.02037.

**Questions:**

Except these weaknesses, I have the following question to inference and experiment.

1. While the proposed method requires few diffusion steps, it does contain many additional steps (line 5 6 7) shown in Algorithm 1. Can the author give a fair comparison of inference time instead of diffusion steps, given that each diffusion step has different cost?

---

> ### Author Response · Authors · 2024-11-22
>
> We thank the reviewer for their constructive feedback.
>
> > Lack an detailed explanation of why the inter-dependency issue is only for discrete diffusion model but not for continuous diffusion model, which can affect the soundness.
>
> We thank the reviewer for bringing up this important question. Intuitively, the inter-dependency problem is more severe in discrete diffusion models compared to continuous diffusion models because in discrete diffusion models, we have to *change a small number of variables by a lot* in each denoising step, which calls for accurate modeling of the inter-dependency of those variables that are changed, while in continuous diffusion models, all variables are *changed by a little* in each denoising step.
>
> More formally, as discussed in Section 3, the inter-dependency issue for discrete diffusion models is caused by the fact that they independently sample each variable from $x_t$ conditioned on $x_{t+1}$. Additionally, as discussed in Appendix D, when the respective losses of many discrete diffusion models are minimized, the models will only be able to learn the ground truth univariate marginals. In the widely adopted absorbing mask noising process, it means that if we unmask multiple tokens at one diffusion step, we will not be able to account for their inter-dependency, hence causing poor sample quality.
>
> In contrast, in continuous diffusion models, we only need to learn the Stein score of the distribution in order to simulate the reverse-time SDE/ODE to draw samples. When the losses of continuous diffusion models are minimized, the models learn the exact Stein score of the distribution, while in discrete diffusion models, the variable inter-dependencies will not be captured even if the losses are minimized.
>
> Some recent continuous diffusion models are able to achieve few-step generation thanks to their ability to learn/construct more straight gradient fields (equivalent to the Stein score up to some simple transformations) between the source and the target distributions. However, this cannot be directly adapted to the discrete world as there is no direct analogy of these gradient fields for discrete data.
>
> > given [1] shows that the SEDD's objective can be exactly derived from ELBO setting. The author also lacks reference for its continuous extension.
>
> Thanks for the suggestion. We agree that the objective of continuous-time discrete diffusion models can be derived from both the ELBO perspective and the likelihood ratio perspective. We have updated the paper to include this as well as related references.
>
> > Lack a clear definition of "copula", while it appears many times in many different ways.
>
> In the paragraph starting at line 229, we adopted the definition of copulas in Sklar (1959), and the extension of copulas to discrete distributions in Geenens (2020), where the information of a copula can be parameterized by odds ratios. Odds ratios are formally defined in Definition 2 and discussed in more details in Appendix C.
>
> > Section 5 is not clear enough
>
> We thank the reviewer for their suggestions. We have incorporated the suggestions and provided more intuitions in each subsection. Specifically, the goal of Section 5.1 is to answer how to approximate $q(X_{t} | x_{t+1})$ using autoregressive models. We show that this can be achieved if the discrete diffusion model uses the absorbing mask noising process. The intuition here is the decomposition of $q(X_{t} | x_{t+1})$ into two distributions (line 292), where the former intuitively captures the joint distribution of generating all masked tokens, and the latter captures only the choice of which currently masked tokens will actually be generated.
>
> After justifying that autoregressive models can be used as the copula model, the goal of Section 5.2 is to further demonstrate how to approximate the I-projection procedure to add back inter-variable dependencies missed by the discrete diffusion model. Specifically, the update step is first provided in Equation (8) before being explained and justified later. Finally, the overall algorithm is described and outlined in Section 5.3 and Algorithm 1.
>
> > Notation/equation should be consistent.
>
> We thank the reviewer for their suggestion. We have updated the notations in Section 5.1 to improve consistency.

---

> > ### Author Response · Authors · 2024-11-22
> >
> > > Ideally, one would like to train an "autoregressive diffusion" model proposed by Emiel Hoogeboom [2], which can make the ordering issue less problematic.
> >
> > As shown in the analysis in Section 4, the copula model does not need to be unbiased to achieve good hybrid models, since the I-projection enforces the combined distribution’s univariate marginals to match exactly with that of the diffusion models, which are much more accurate as they can condition on all unmasked tokens in the sequence. Since the discrete diffusion model does not model the copula at all, for the copula model, any reasonable approximation of the data distribution will provide a net gain to the hybrid model as long as it can provide information about inter-variable dependencies.
> >
> > Although the I-projection is not perfectly solved for autoregressive copula models, we observe that (i) according to Theorem 1, the I-projection procedure is “simple” as it can be treated as a convex optimization problem, and (ii) empirical results demonstrate that the hybrid model achieves better performance than its base models across many tasks, which suggests the empirical effectiveness of DCD.
> >
> > > Can the author give a fair comparison of inference time instead of diffusion steps, given that each diffusion step has different cost?
> >
> > We have conducted additional experiments in unconditional text generation to better understand the relation between quality (measured by generative perplexity), diversity (measured by sentence entropy), and speed for DCD and its baselines. Specifically, we run the more efficient version of DCD described in the last paragraph of Section 5.3 and Appendix E to generate text sequences of lengths 128 and 1024. In addition to SEDD and GPT2, the two base models used by DCD, we compare them with MDLM, a more recent discrete diffusion model that is more efficient than SEDD. Note that DCD can use any discrete diffusion model as its base model.
> >
> > Below are tables containing results for sequence length 128. For better visualization, we have plotted the results for both length 128 and length 1024 in Appendix F of the revised paper. First, we compare the sample time and the generative perplexity (the second and the fourth sub-plot in Figure 7). Compared to SEDD, GPT, and MDLM, DCD consistently achieves better generative perplexity given a fixed runtime constraint. It also requires less time to achieve a desired perplexity value.
> >
> > | # steps         | 2       | 4       | 8       | 16       | 32       | 64       | 128       | 256       |
> > |----------------|---------|---------|---------|---------|---------|---------|---------|---------|
> > | SEDD Entropy   | 4.4874  | 4.4046  | 4.3330  | 4.2647  | 4.1895  | 4.1223  | 4.0761  | 4.0084  |
> > | SEDD Perplexity| 2958.24 | 892.49  | 345.83  | 167.98  | 104.09  | 77.79   | 67.05   | 59.13   |
> > | SEDD Runtime| 0.0102 | 0.0171 | 0.0315 | 0.0587 | 0.1148 | 0.1858 | 0.3684 | 0.7445 |
> > |MDLM Entropy| 4.5002 | 4.4399 | 4.3562 | 4.2262 | 4.0507 | 3.7471 | 3.6229 | 3.5217 |
> > |MDLM Perplexity| 3531.30 | 1373.38 | 503.03 | 202.65 | 117.93 | 66.36 | 48.13 | 36.95 |
> > |MDLM Runtime| 0.0116 | 0.0164 | 0.0229 | 0.0413 | 0.0698 | 0.1280 | 0.2557 | 0.4766 |
> > |DCD Entropy | 4.2677 | 4.1824 | 4.0976 | 4.0297 | 4.0099 | - | - | - |
> > |DCD Perplexity| 73.48 | 58.73 | 43.95 | 30.15 | 20.87 | - | - | - |
> > |DCD Runtime| 0.0582 | 0.0671 | 0.0862 | 0.1238 | 0.2038 | - | - | - |
> >
> > Additionally, we compare the perplexity and diversity of the generated text sequences. Following community standards, we adopt the sentence entropy to measure the diversity of generated text. Specifically, the entropy of each text sequence is the entropy of its token frequency distribution, and the final sentence entropy is the average entropy over all generated sequences. The desired behavior is to have low generative perplexity and high sentence entropy (which means high diversity). Results are shown in the table below and Figure 7's first and third sub-plot. Compared to the two discrete diffusion models (SEDD and MDLM), DCD achieves better generative perplexity under the same entropy, which offers a better perplexity-diversity tradeoff. Compared to the autoregressive GPT model, although the entropy of DCD is lower, it achieves better generative perplexity with slightly worse entropy.
> >
> > (To be continued in the next comment.)

---

> > > ### Author Response · Authors · 2024-11-22
> > >
> > > Conceptually, the efficiency of DCD is discussed in Section 6.2 and Figure 5. By using the autoregressive order for sequence infilling, for DCD with N diffusion steps, 2*N calls of the diffusion model are needed (line 6 in Algorithm 1). Additionally, there is a constant cost of running the autoregressive model to sample all tokens once (since we can use KV-caching; described in line 396). Therefore, as shown in Figure 5, the runtime of DCD with N diffusion steps is 2 times of the base diffusion model plus a constant overhead from the autoregressive model, which is amortized as the number of diffusion steps goes up. In comparison with the 8 to 32 times saving in number of denoising steps, the computation cost is only 2 to 4 times larger per denoising step.
> > >
> > > We note that this version of DCD changes the ordering for unmasking tokens, which can be a limitation for certain tasks (e.g., when we desire random unmasking). As described in the paper, this variant is also used in the text infilling and antibody infilling tasks, so this does not limit the model’s ability to do infilling/imputation, but it could affect certain applications such as controlling the generation process with techniques similar to classifier-based/-free guidance. However, the additional results demonstrate that DCD can efficiently generate high-quality samples. They could also motivate future work to further improve DCD or to solve the inter-variable dependency problem of discrete diffusion models in a more principled way.

---

> > > > ### Author Response · Authors · 2024-11-27
> > > >
> > > > Dear reviewer nyBv,
> > > >
> > > > Thank you again for the insightful comments. Since the discussion period is about to end, could you kindly check if our response addressed your concerns? We are happy to answer any follow-up questions and concerns.
> > > >
> > > > Best,
> > > >
> > > > Authors

---

### Meta-Review · Area_Chair_mQ4Y · 2024-12-22

**Metareview:**

This paper introduces Discrete Copula Diffusion (DCD), a framework to improve discrete diffusion models by addressing their inability to capture inter-variable dependencies at each denoising step. The method integrates a discrete diffusion model with an autoregressive copula model to achieve higher-quality sample generation with significantly fewer denoising steps. Theoretical analysis is provided to support the proposed approach, and experiments demonstrate its effectiveness in text generation and other tasks.

**Additional Comments On Reviewer Discussion:**

The reviewers found the proposed approach to be a novel and theoretically sound contribution. However, significant concerns were raised about the computational efficiency of the method, as it involves substantial overhead per denoising step by sampling from an autoregressive model.. They also noted that the reliance on autoregressive copula models introduces bias. The authors provided a detailed response that showed improved runtime and quality trade-offs over recent methods like SEDD and MDLM, which arise from the use of a KV caching mechanism that was explained in additional detail in the rebuttal. The authors also demonstrated the method’s applicability to longer sequences. Post-rebuttal, the reviewers remained divided, with some acknowledging the value of the research direction but others still concerned about the clarity of the writing of the paper. We strongly suggest incorporating speed results from the rebuttal into the main paper, adding autoregressive baselines to every table, reporting performance for both versions of DCD and/or making it clear which method is used in which experiment, reporting parameter counts for each model (given that the improved performance could well be explained by increasing the parameter count via using a second pre-trained AR model). Lastly, we suggest improving the clarity to better explain the KV-caching mechanism and the left-to-right formulation of DCD (currently, presented very succinctly at the end of 5.3).

---

### Decision · Program_Chairs · 2025-01-22

Accept (Poster)